



# Revisiting the vapor diffusion coefficient in dry snow

Andrew Hansen[Professor Emeritus, University of Wyoming]

[1]Dept. of Mechanical Engineering, Laramie, WY 82071

**Correspondence:** A. C. Hansen (hansen@uwyo.edu)

**Abstract.** A substantial degree of uncertainty surrounds the value of the diffusion coefficient for water vapor diffusing through snow under the influence of a temperature gradient. A collection of theoretical, numerical, and experimental studies suggest the value of the normalized diffusion coefficient of snow with respect to water vapor diffusing through humid air ranges from near zero to as high as 7. The challenges in quantifying the diffusion coefficient are attributed to the fact that snow is a phase changing mixture of ice and humid air. Phase changes involving sublimation and condensation of water molecules significantly alter diffusion paths that water vapor molecules must travel through in the complex ice/humid air microstructure.

I identify 4 major diffusion mechanisms caused by the introduction of an ice phase into humid air that should be accounted for in any calculation of the diffusion coefficient. Mathematical and experimentally motivated arguments enable one to estimate the physical significance of each of these diffusion mechanisms. Using this information, three separate models of increasing complexity are developed to provide estimates of the diffusion coefficient for snow. Finally, several prior studies are anchored to a formal definition of the diffusion coefficient and, if necessary, adjusted to account for the 4 diffusion mechanisms presented. The end result is a remarkable consistency in predicted values for vapor diffusion in snow, showing the normalized diffusion coefficient with respect to water vapor in air to be slightly enhanced at all densities, with typical values between 1 and 1.3. These values are consistent with the models developed herein.

## 1   History

A colorful history of mystery and intrigue surrounds the value of the diffusion coefficient of water vapor in snow. The principal reason behind this enigma is that snow is a phase changing mixture of ice and humid air. Hence, diffusion paths of water molecules may involve phase changes due to sublimation or condensation while also following the tortuous route through the humid air.

The challenge of quantifying the diffusion coefficient has been addressed through experimental, theoretical, and numerical studies. Yosida (1955) performed pioneering experimental studies and concluded the diffusion coefficient for snow was 4-5 times larger than that for water vapor diffusing through air. Yosida correctly identified ice blocking pathways as a mechanism impeding diffusion in snow. To counter this effect, Yosida introduced the idea of a "hand-to-hand" diffusion process whereby





water molecules sublimate off the tops of ice grains whose local temperatures are warmer than the surrounding air. After traveling through the pore space of humid air, driven by a local temperature gradient, they then condense on to the lower surface of an ice grain whose local temperature is cooler than that of air. The net effect is that ice grains act as an instantaneous source and sink of water vapor, thereby shortening the diffusion path of a water molecule. A plethora of experimental observations at

the microscale confirm this brilliant analogy.

Sommerfeld et al. (1987) performed interesting experiments to measure mass transfer in snow by monitoring changes in naturally occurring isotopes when a layer of snow gains or loses mass. Results indicate the diffusion coefficient for snow is approximately twice that of water vapor diffusing in air.

Giddings and LaChapelle (1962) argued for a vapor diffusion coefficient for snow to be less than that for humid air, owing

to the increased tortuosity of diffusion paths. Sokratov and Maeno (2000) performed experiments supporting this view.

Colbeck (1993) attacked the diffusion problem for snow from a theoretical standpoint by studying a particle-to-particle model of vapor diffusion. Colbecks's results suggested a snow diffusion coefficient 4-7 times higher than that of water vapor diffusing through air.

Foslien (1994) developed a model for diffusion of water vapor in snow producing normalized values of the diffusion coef-

ficient with respect to diffusion in humid air from 1-1.23 over a broad range of densities. Foslien's model is based on a heat transfer analysis for two simplified microstructures believed to represent the extreme possibilities for heat transfer in a humid air/ice microstructure. He then utilized arguments from quantitative stereology to construct a model for snow. The diffusion coefficient falls out as a natural result of this one-dimensional heat transfer analysis.

Christon (1990) performed some of the first numerical studies on heat and mass transfer in snow, solving a fully coupled

heat and mass transfer problem on several unit cell models of an ice lattice. Christon's results suggest a normalized diffusion coefficient for snow between 1 and 1.93.

The advent of X-ray tomography of snow ushered in an exciting new era in snow science by opening a window into monitoring microstructural evolution while also quantifying precise representative volume elements (RVE's) for use in numerical heat and mass transfer studies. Pinzer et al. (2012) used finite element analysis of a RVE to predict the mass flux of water

vapor in snow. Results from 3 different snow samples with densities ranging from $257 - 312 \, \mathrm{kg \, m^{-3}}$ show there is minimal enhancement in the mass flux in the humid air phase of snow compared to that for humid air alone, suggesting a normalized diffusion coefficient in the range 1.05-1.13.

Calonne et al. (2014) performed numerical analyses on snow RVE's similar to that of Pinzer et al. (2012) to determine the diffusion coefficient tensor for snow. They analyzed snow over a broad range of densities from approximately $100 -$

$500 \, \mathrm{kg \, m^{-3}}$. Notably, their results always produced a normalized diffusion coefficient less than 1 whose value decreased linearly with increasing density. The mechanisms contributing to retarding vapor diffusion are attributed to the ice phase acting as a blockage for diffusion paths and the tortuosity of diffusion paths.

In summary, a broad view of the history of research on mass transfer in snow suggests there is no definitive answer to the value of the vapor diffusion coefficient in snow. A partial explanation for many of the noted discrepancies may lie in a lack



of a precise and consistent definition of the vapor diffusion coefficient. For instance, the numerical studies of Christon (1990), Pinzer et al. (2012), and Calonne et al. (2014) all use different methods to evaluate the mass flux and/or the diffusion coefficient.

In this paper, a formal definition of the diffusion coefficient for snow is developed based on arguments from continuum mechanics at the macroscale. This definition is further grounded to the development of energy and mass balance equations for

snow that account for mass exchange between the humid air and ice phase. Next, 4 major diffusion mechanisms are identified that should be accounted for in any calculation of the diffusion coefficient. Moreover, rigorous physical arguments are presented that enable one to estimate the significance of each of these diffusion mechanisms. The end result leads to 3 analytical models of increasing complexity for the vapor diffusion coefficient.

Finally, several prior studies are anchored to the formal definition of the diffusion coefficient and, if necessary, adjusted to

account for the 4 diffusion mechanisms presented. The end result is a remarkable consistency in predicted values for vapor diffusion in snow, showing the normalized diffusion coefficient with respect to water vapor in air to be slightly enhanced with typical values between 1 and 1.3. These values are consistent with the models developed herein.

## 2   Mass transfer in snow

The critical heat and mass transfer mechanisms for snow metamorphism play out at two distinctly different geometric and

time scales. At the microscale (on the order of millimeters), snow exhibits an extremely complex and evolving microstructure consisting of ice grains and humid air. At the macroscale, snow is considered to be a phase changing mixture of ice and humid air. The humid air itself may be considered to be an ideal gas mixture of water vapor and air. A schematic of these two scales is shown in Figure 1. The individual ice grains and surrounding humid air identified in the circular region represent the microscale. At the macroscale, this ice/humid air microstructure is homogenized into a continuum mixture by apprrpriate

volume averaging of microscale properties.

The following subscripts are used to define variables throughout:

–   (s) $\sim$ snow as well as other simplified ice/humid air microstructures,

–   (i) $\sim$ ice constituent,

–   (ha) $\sim$ humid air constituent, and

–   (v) $\sim$ water vapor constituent.

For a snow mixture, the volume fractions of ice and humid air are space filling, leading to the relation

$$\phi_{\mathrm{i}} + \phi_{\mathrm{ha}} = 1, \tag{1}$$

where $\phi_{\alpha}$ represents the volume fraction of constituent $\alpha$. The density of snow is defined by the relation

$$\rho = \phi_{\mathrm{i}}\gamma_{\mathrm{i}} + \phi_{\mathrm{ha}}\gamma_{\mathrm{ha}}, \tag{2}$$





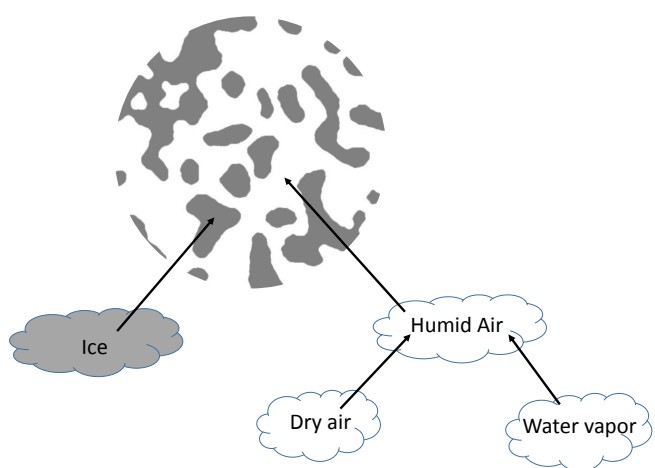

**Figure 1.** Schematic showing a continuum point of snow with the associated constituents

where $\gamma_\alpha$ represents the true density of constituent $\alpha$. $\gamma_\alpha$ is used to represent the true constituent density as, in classical mixture theories, $\rho_\alpha$ typically denotes a dispersed constituent density defined by the mass of constituent $\alpha$ per volume of the mixture. Hence, in terms of dispersed densities, the density of snow may be expressed as

$$\rho = \rho_i + \rho_{ha}. \tag{3}$$

The explicit presence of constituent volume fractions in the definition of snow density found in Eq. (2) is important for two reasons. First, the constituent densities have their normal physical meanings. Second, in macroscale heat and mass transfer studies of snow, one is often interested in tracking changes in the ice volume fraction due to mass transfer. The explicit presence of the volume fraction affords us the opportunity to do so.

The humid air is represented as a miscible mixture where both the water vapor and air occupy the same volume leading to

$\phi_v = \phi_{ha}.$

In this special case, the dispersed vapor density, $\rho_v$, and the true vapor density, $\gamma_v$, are identical.

An important aspect of placing the vapor diffusion coefficient on a consistent footing is anchoring the definition to the balance equations for a phase changing mixture. In framing the macroscale balance of mass and energy for snow, the following assumptions are utilized:

– convection is neglected. Convection only occurs in extreme weather conditions such as near the top of a snowpack in the presence of a strong wind or extremely large temperature gradients. Foslien (1994) provides support for this assumption through the calculation of a Rayleigh number for porous media. His results show the Rayleigh for snow is an order of magnitude below the number required for the onset of convection. Additional support for neglecting convection at this scale is found in Kaempfer et al. (2005) and Hammonds et al. (2015).





It should be noted that convection effects may be significant at the much larger scales found in the field as has been documented by Sturm and Johnson (1991) and Sturm and Benson (1997).

– radiative heat transfer is neglected. Radiation may be important when studying near surface phenomena such as the development of surface hoar.

– the humid air phase is assumed saturated at all times. Justification for this assumption is provided in Hansen and Foslien (2015).

– vapor transport is assumed to be diffusion limited,

– snow is assumed to be isotropic—an important feature of the present development although this constraint may be relaxed,

– the temperatures of the ice and humid air at the *macroscale* are identical permitting one to write

$$\theta = \theta_{\mathrm{ha}} = \theta_{\mathrm{i}}.$$

Justification for this assumption is found in Hansen and Foslien (2015) where the time scale for heat conduction in a snow pack to the time scale for heat conduction at the microscale is on the order of $10^6$. Note that a very different situation exists at the microscale where the temperature of ice and humid air are different. Indeed, different microscale
constituent temperatures are a requirement for sublimation and condensation of water molecules to occur.

## 2.1 Macroscale: A precise definition of the vapor diffusion coefficient

Consider a body $\mathcal{B}$ of snow with a differential surface area, $dS$, as shown in Figure 2.

Across the surface $dS$, energy and mass flux vectors are identified as

– $\boldsymbol{q}_{\mathrm{s}}$ is the energy flux given by,

$$\boldsymbol{q}_{\mathrm{s}} = \boldsymbol{q}^{\mathrm{c}} + \boldsymbol{q}^{\mathrm{d}},$$

where $\boldsymbol{q}^{\mathrm{c}}$ is the conductive flux and $\boldsymbol{q}^{\mathrm{d}}$ represents heat transfer due to water wapor diffusion (Bird and Lightfoot (1960)),

– $\boldsymbol{j}_{\mathrm{s}}$ is the mass flux due to quiet diffusion of water vapor though the humid air phase.

The heat flux is defined by Fourier's law of heat conduction as

$$\boldsymbol{q}^{\mathrm{c}} = -k_{\mathrm{s}} \boldsymbol{\nabla} \theta, \tag{4}$$

where $k_{\mathrm{s}}$ is the thermal conductivity of snow and $\boldsymbol{\nabla}\theta$ is the temperature gradient.

Let $\gamma_{\mathrm{v}}$ and $\boldsymbol{v}_{\mathrm{v}}$ denote the density and velocity of water vapor in humid air, respectively. The mass flux through humid air is given by

$$\boldsymbol{j}_{\mathrm{v}} = \gamma_{\mathrm{v}} \boldsymbol{v}_{\mathrm{v}}. \tag{5}$$





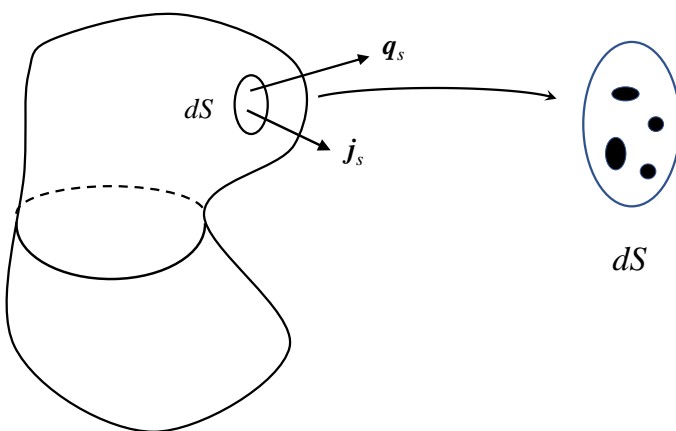

**Figure 2.** A continuum body of snow showing mass and energy flux vectors emanating from a differential surface area, $dS$.

In the case of snow, two important features of Eq. (5) are altered. First, an obvious difference is that the differential surface area of the humid air is reduced by the presence of the ice. Hence, one can write

$$dS^{\text{eff}} = \left( \frac{dS_{\text{ha}}}{dS} \right) dS = \phi_{\text{ha}} dS, \tag{6}$$

where the well known result from stereology is utilized that the volume fraction, area fraction, and lineal fraction are equal.

A much more subtle aspect of water vapor diffusion in snow is that diffusion paths are altered by the presence of the ice. Hand-to-hand exchange of water vapor shortens diffusion paths while tortuosity increases the diffusion path. The net effect of the altered diffusion paths is that the time for water vapor to traverse snow under a temperature gradient is different than the time for diffusion occurring in humid air alone. The influence of a different time scale is manifest in the diffusive flux which may be described as mass transport of water vapor crossing a surface per unit of area per unit of time.

To account for this altered time scale for water vapor diffusion, the notion of intrinsic time, $t^*$, is introduced abstractly. In its simplest terms, intrinsic time is the apparent time for a water vapor molecule to traverse an RVE versus the time a water vapor molecule would move through humid air alone. The word "apparent" is necessary in that water vapor goes through sublimation and diffusion and hence it is impossible to track a single water molecule.

An intrinsic diffusion velocity is also introduced as $\boldsymbol{v}_{\text{v}}^*$ and represents the diffusion velocity referenced to intrinsic time.

Importantly, intrinsic time and the intrinsic diffusion velocity may be quantified for the complex ice/humid air microstructure found in snow.

Noting these two features of water vapor diffusion in snow, the mass flux for snow is expressed as

$$\boldsymbol{j}_{\text{s}} = \phi_{\text{ha}} \gamma_{\text{v}} \boldsymbol{v}_{\text{v}}^*. \tag{7}$$





In an analogous form to Fourier's law of heat conduction, the mass flux for snow is defined by Fick's law and expressed as (Bird and Lightfoot (1960))

$$\boldsymbol{j}_{\mathrm{s}} = -\gamma_{\mathrm{ha}} D_{\mathrm{s}} \boldsymbol{\nabla} \left( \frac{\gamma_{\mathrm{v}}}{\gamma_{\mathrm{ha}}} \right), \tag{8}$$

where $D_{\mathrm{s}}$ is the diffusion coefficient for snow. The challenge is met immediately as a host of diffusion mechanisms are buried within $D_{\mathrm{s}}$. As with the mass flux, an important feature of $D_{\mathrm{s}}$ is the presence of time in the units of this coefficient.

The diffusive flux can be expanded to give

$$\boldsymbol{j}_{\mathrm{s}} = -D_{\mathrm{s}} \boldsymbol{\nabla} \gamma_{\mathrm{v}} + \frac{\gamma_{\mathrm{v}}}{\gamma_{\mathrm{ha}}} D_{\mathrm{s}} \boldsymbol{\nabla} \gamma_{\mathrm{ha}}, \tag{9}$$

but the second term on the right is negligibly small because the mass fraction of saturated water vapor in air at 273 K is about $4(10)^{-3}$. Hence, mass transfer of water vapor at the macroscale may be described by

$$\boldsymbol{j}_{\mathrm{s}} = -D_{\mathrm{s}} \boldsymbol{\nabla} \gamma_{\mathrm{v}}. \tag{10}$$

Because the vapor density is saturated, it may be expressed as purely a function of temperature (Dorsey, 1968) leading to

$$\boldsymbol{\nabla} \gamma_{\mathrm{v}} = \frac{\mathrm{d}\gamma_{\mathrm{v}}}{\mathrm{d}\theta} \boldsymbol{\nabla} \theta \qquad \text{and} \qquad \frac{\partial \gamma_{\mathrm{v}}}{\partial t} = \frac{\mathrm{d}\gamma_{\mathrm{v}}}{\mathrm{d}\theta} \frac{\partial \theta}{\partial t}.$$

Noting the above, the mass flux for snow may be expressed as

$$\boldsymbol{j}_{\mathrm{s}} = \phi_{\mathrm{v}} \gamma_{\mathrm{v}} \boldsymbol{v}_{\mathrm{v}}^{*} = -D_{\mathrm{s}} \left( \frac{d\gamma_{\mathrm{v}}}{d\theta} \right) \boldsymbol{\nabla} \theta. \tag{11}$$

Equation (11) provides a firm grounding for a precise definition for mass flux and the associated vapor diffusion coefficient for snow. In Section 3.2, the intrinsic diffusion velocity, $\boldsymbol{v}_{\mathrm{v}}^{*}$, for snow is explicitly quantified. Attention is also drawn to the macroscale temperature gradient of snow, $\boldsymbol{\nabla}\theta$, as an important aspect of anchoring the definition of the vapor diffusion coefficient. Specifically, in Eq. (11), $\boldsymbol{\nabla}\theta$ represents the macroscale temperature gradient of snow and not the temperature gradient of the humid air where diffusion is actually occurring.

In addition to the precise definition of $D_{\mathrm{s}}$ outlined in Eq. (11), it is useful to frame the vapor diffusion coefficient in the context of the balance equations for mass and energy of snow. Following the development of Foslien (1994) and summarized in Hansen and Foslien (2015), a single equation for the macroscale temperature $\theta(x,t)$ assumes the form

$$\left( \phi_{\mathrm{ha}} \gamma_{\mathrm{ha}} C_{\mathrm{ha}}^{V} + \phi_{\mathrm{i}} \gamma_{\mathrm{i}} C_{\mathrm{i}}^{V} + u_{\mathrm{sg}} \phi_{\mathrm{ha}} \frac{d\gamma_{\mathrm{v}}}{d\theta} \right) \frac{\partial \theta}{\partial t} = \boldsymbol{\nabla} \cdot \left( k_{\mathrm{s}} + u_{\mathrm{sg}} D_{\mathrm{s}} \frac{d\gamma_{\mathrm{v}}}{d\theta} \right) \boldsymbol{\nabla} \theta, \tag{12}$$

where, for constituent $\alpha$ representing ice or humid air:

 – $\phi_{\alpha}$ is the volume fraction of constituent $\alpha$,

 – $\gamma_{\alpha}$ is the density of constituent $\alpha$,

 – $C_{\alpha}^{V}$ is the specific heat at constant volume of constituent $\alpha$,





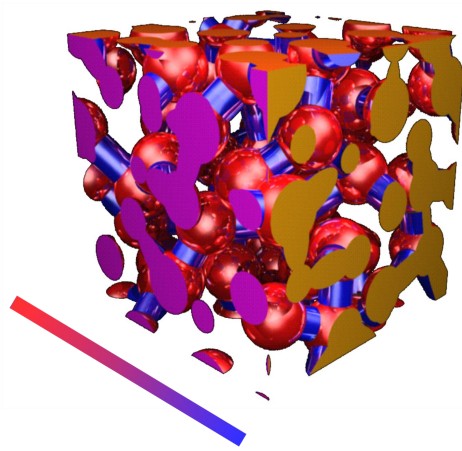

**Figure 3.** Idealized snow microstructure of a RVE with ice grains (red) and grain bonds (blue). Temperature bar indicates warm and cold directions.

    – $k_\mathrm{s}$ is the thermal conductivity of snow, excluding latent heat transfer,

    – $D_\mathrm{s}$ is the diffusion coefficient of snow, and

    – $u_\mathrm{sg}$ is the latent heat of sublimation of ice.

    Eq. (12) was also derived by Calonne et al. (2014) by homogenization of multiple scale expansions. Both the work of Foslien

(1994) and Calonne et al. (2014) are grounded in the physics of heat and mass transfer at the microscale.

    The energy flux accounting for heat conduction combined with energy transfer due to water vapor diffusion in Eq. (12) may be identified as

$$\boldsymbol{q}_\mathrm{s} = -\left(k_\mathrm{s} + u_\mathrm{sg} D_\mathrm{s}\frac{\mathrm{d}\gamma_\mathrm{v}}{\mathrm{d}\theta}\right)\boldsymbol{\nabla}\theta. \tag{13}$$

The energy flux provides additional valuable context for the definition of the diffusion coefficient at the macroscale.

## 2.2  Microscale: RVE analysis

Consider a RVE of an idealized snow microstructure as shown in Figure 3. Gold regions on the front face indicate ice where blockage of vapor diffusion will occur across the surface. Relative to the RVE, let $\boldsymbol{\xi}$ represent spatial coordinates. For clarity, the dependence of $\boldsymbol{\xi}$ is always explicitly written for any spatially varying dependent variable at the microscale, e.g., $\theta(\boldsymbol{\xi},t)$ represents the microscale temperature field, whereas $\theta$ represents the macroscale temperature.

Vapor diffusion in snow is driven by temperature gradients and, therefore, it is useful to examine relations between macroscale and microscale temperature gradients. To begin, the macroscale temperature gradient for snow is the volume average of the





local microscale temperature gradients and may be expressed as

$$\boldsymbol{\nabla}\theta = \frac{1}{V}\int_V \boldsymbol{\nabla}_\xi \theta\left(\boldsymbol{\xi},t\right)\mathrm{d}V. \tag{14}$$

The subscript $\xi$ on the gradient operator in Eq. (14) is used to emphasize the gradient applies at the microscale.

Breaking the volume average into separate domains for the ice and humid air constituents leads to

$$5 \quad \boldsymbol{\nabla}\theta = \phi_\mathrm{i}\left(\frac{1}{V_\mathrm{i}}\int_{V_\mathrm{i}} \boldsymbol{\nabla}_\xi \theta\left(\boldsymbol{\xi},t\right)\mathrm{d}V\right) + \phi_\mathrm{ha}\left(\frac{1}{V_\mathrm{ha}}\int_{V_\mathrm{ha}} \boldsymbol{\nabla}_\xi \theta\left(\boldsymbol{\xi},t\right)\mathrm{d}V\right), \tag{15}$$

or

$$\boldsymbol{\nabla}\theta = \phi_\mathrm{i}\left(\boldsymbol{\nabla}\theta\right)_\mathrm{i} + \phi_\mathrm{ha}\left(\boldsymbol{\nabla}\theta\right)_\mathrm{ha}, \tag{16}$$

where $\left(\boldsymbol{\nabla}\theta\right)_\mathrm{i}$ and $\left(\boldsymbol{\nabla}\theta\right)_\mathrm{ha}$ are volume averaged *macroscale* constituent temperature gradients.

Now consider the mass flux of water vapor in the humid air phase at the *microscale* written as

$$10 \quad \boldsymbol{j}_\mathrm{v}(\boldsymbol{\xi}) = \gamma_\mathrm{v}(\boldsymbol{\xi})\boldsymbol{v}_\mathrm{v}(\boldsymbol{\xi}) = -D_\mathrm{v\text{-}a}\left(\frac{d\gamma_\mathrm{v}(\boldsymbol{\xi})}{d\theta}\right)\boldsymbol{\nabla}_\xi \theta(\boldsymbol{\xi}), \tag{17}$$

where $D_\mathrm{v\text{-}a}$ represents the binary diffusion coefficient of water vapor in air. Note that $\boldsymbol{j}_\mathrm{v}$ is used to represent a microscale flux while $\boldsymbol{j}_\mathrm{s}$ is referenced to the macroscale. Volume averaging Eq. (17) over the humid air phase gives

$$\overline{\gamma_\mathrm{v}(\boldsymbol{\xi})\boldsymbol{v}_\mathrm{v}(\boldsymbol{\xi})} = \overline{-D_\mathrm{v\text{-}a}\left(\frac{d\gamma_\mathrm{v}(\boldsymbol{\xi})}{d\theta}\right)\boldsymbol{\nabla}_\xi \theta(\boldsymbol{\xi})}, \tag{18}$$

where an overbar represents the volume average of the *humid air* phase. The local temperature differences are extremely small in a RVE, allowing one to treat $\gamma_\mathrm{v}$, $\frac{d\gamma_\mathrm{v}}{d\theta}$, and $D_\mathrm{v\text{-}a}$ as constant, leading to

$$\gamma_\mathrm{v}\overline{\boldsymbol{v}_\mathrm{v}(\boldsymbol{\xi})} = -D_\mathrm{v\text{-}a}\left(\frac{d\gamma_\mathrm{v}}{d\theta}\right)\overline{\boldsymbol{\nabla}_\xi \theta(\boldsymbol{\xi})}. \tag{19}$$

Equation (19) shows that the volume averaged humid air diffusion velocity is a linear function of the volume averaged humid air temperature gradient. This relationship is important as, while temperature gradients may be very large in some locations at the microscale, it is the average humid air temperature gradient that drives macroscale vapor diffusion. Note that Eq. (19) is, in effect, a macroscale relation as a consequence of volume averaging. Hence, in terms of macroscale variables, Eq. (19) becomes

$$\gamma_\mathrm{v}\boldsymbol{v}_\mathrm{v} = -D_\mathrm{v\text{-}a}\left(\frac{d\gamma_\mathrm{v}}{d\theta}\right)\left(\boldsymbol{\nabla}\theta\right)_\mathrm{ha}. \tag{20}$$

Finally, consider the normalized value of the vapor diffusion coefficient for snow with respect to the binary diffusion coefficient of water vapor in air given by $(D_\mathrm{s}/D_\mathrm{v\text{-}a})$. Note that, in the limit as the ice phase goes to zero, this quantity tends to one, i.e.,

$$\lim_{\phi_\mathrm{i}\to 0}\left(\frac{D_\mathrm{s}}{D_\mathrm{v\text{-}a}}\right) = 1. \tag{21}$$

As the ice volume fraction increases from zero, there are 4 major factors that either enhance or reduce the normalized diffusion coefficient including:





1. The ice phase acts to block diffusion paths, thereby impeding diffusion.

2. The ice phase acts to shorten diffusion paths through the hand-to-hand diffusion mechanism described by Yosida (1955), thereby enhancing diffusion.

3. Elevated temperature gradients in the humid air phase of snow enhance diffusion. The temperature gradient effect is manifest in the transition from the microscale humid air temperature gradient to the macroscale temperature gradient for snow.

4. Tortuosity of diffusion paths drive the normalized diffusion coefficient lower.

In developing a diffusion model for snow, I begin by studying diffusion at the microscale and address these 4 factors noted in the transition to a macroscale vapor diffusion coefficient for snow.

## 3 Vapor diffusion coefficient models

In this section, 3 diffusion coefficient models for mixtures of ice and humid air are developed, aimed at increasingly accurate estimates for the vapor diffusion coefficient in snow.

### 3.1 An upper bound for vapor diffusion

Consider water vapor transport in a layered ice/humid air microstructure as shown in Figure 4. Although the ice phase appears to choke off diffusion of water vapor entirely, the opposite is true, as the ice acts as an instantaneous source and sink for water vapor, thereby increasing diffusion rates. Field and laboratory studies by Greene (2007) and Hammonds et al. (2015), respectively, confirm this source/sink phenomenon. Specifically, an ice crust in snow subjected to a temperature gradient, resulting from a warm ground surface and a colder air surface, shows deposition occurring on the lower (warmer) ice surface and sublimation on the upper (cooler) surface.

The layered microstructure is believed to represent an upper bound for water vapor diffusion in snow for the following reasons:

– diffusion paths are straight, thereby eliminating tortuosity as a factor,

– the ice phase shortens all diffusion paths across the unit cell of Figure 4(b) to $L_{\mathrm{ha}}$ as opposed to $L_{\mathrm{T}}$,

– the temperature gradient of the ice phase is approximately zero compared to the humid air temperature gradient, owing to the discontinuous nature of the ice phase and the fact that the thermal conductivity of ice is nearly 100 times larger than that of humid air. This feature implies the volume averaged temperature gradient in the humid air is the largest possible compared to the volume averaged humid air temperature gradient in any other ice/humid air microstructure that is representative of snow. The volume averaged humid air temperature gradient drives the macroscale diffusion velocity.



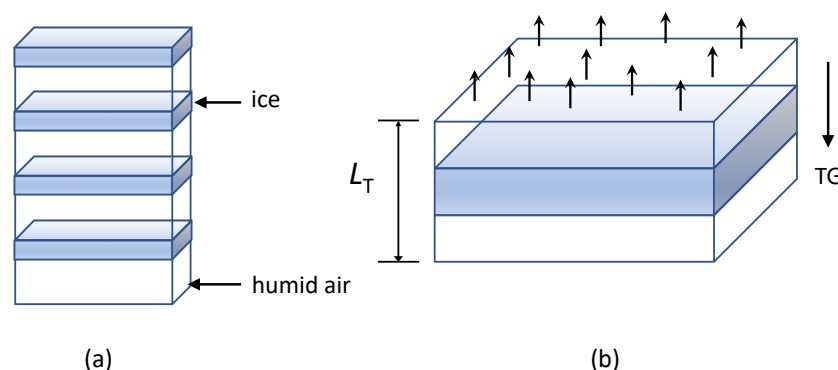

**Figure 4.** (a) Layered ice/humid air microstructure (b) Unit cell showing diffusion off the top surface

Consider a diffusion path traversing the unit cell of Figure 4(b). Because the ice phase acts as a source and sink for water vapor, the intrinsic time necessary for water vapor to cross the unit cell is

$$t^* = L_{\mathrm{ha}}/v_{\mathrm{v}}. \tag{22}$$

In contrast, the time required for water vapor to traverse a unit cell comprised entirely of humid air is given by

$$5 \quad t = L_{\mathrm{T}}/v_{\mathrm{v}}, \tag{23}$$

where $L_{\mathrm{T}}$ represents the total distance across the unit cell.

Dividing Eq. (22) by Eq. (23) shows the intrinsic time for vapor transport in the layered microstructure is

$$t^*/t = L_{\mathrm{ha}}/L_{\mathrm{T}}$$

or

$$10 \quad t^* = \phi_{\mathrm{ha}}t \tag{24}$$

The intrinsic velocity is related to the true diffusion velocity by

$$v_{\mathrm{v}}^* = L_{\mathrm{T}}/t^*$$
$$= v_{\mathrm{v}}/\phi_{\mathrm{ha}}. \tag{25}$$

For instance, for a humid air volume fraction of 0.5, the intrinsic (apparent) velocity is twice the actual velocity.





Now consider the local mass flux volume averaged over the humid air domain as given by Eq. (20). If the averaging process for mass flux is extended over the entire domain (ice and humid air), there follows for a 1-D temperature gradient

$$\phi_{\mathrm{ha}}\gamma_{\mathrm{v}}v_{\mathrm{v}} = -\phi_{\mathrm{ha}}D_{\mathrm{v\text{-}a}}\left(\frac{d\gamma_{\mathrm{v}}}{d\theta}\right)\left(\frac{\partial\theta}{\partial x}\right)_{\mathrm{ha}}. \tag{26}$$

Dividing Eq. (26) by $\phi_{ha}$ and introducing the intrinsic velocity of Eq. (25) gives

$$\phi_{\mathrm{ha}}\gamma_{\mathrm{v}}v_{\mathrm{v}}^{*} = -D_{\mathrm{v\text{-}a}}\left(\frac{d\gamma_{\mathrm{v}}}{d\theta}\right)\left(\frac{\partial\theta}{\partial x}\right)_{\mathrm{ha}}. \tag{27}$$

From Eq. (11), the one-dimensional form of the mass flux for the mixture is given by

$$j_{\mathrm{s}} = \phi_{\mathrm{v}}\gamma_{\mathrm{v}}v_{\mathrm{v}}^{*} = -D_{\mathrm{s}}\left(\frac{d\gamma_{\mathrm{v}}}{d\theta}\right)\left(\frac{\partial\theta}{\partial x}\right). \tag{28}$$

Comparing Eqs. (27) and (28) leads to

$$D_{\mathrm{s}}\left(\frac{\partial\theta}{\partial x}\right) = D_{\mathrm{v\text{-}a}}\left(\frac{\partial\theta}{\partial x}\right)_{\mathrm{ha}}. \tag{29}$$

Now recall from Eq. (16), the mixture and constituent temperature gradients are related by

$$\frac{\partial\theta}{\partial x} = \phi_{\mathrm{i}}\left(\frac{\partial\theta}{\partial x}\right)_{\mathrm{i}} + \phi_{\mathrm{ha}}\left(\frac{\partial\theta}{\partial x}\right)_{\mathrm{ha}}. \tag{30}$$

Noting $\left(\frac{\partial\theta}{\partial x}\right)_{\mathrm{i}} \approx 0$, Eq. (29) becomes

$$D_{\mathrm{s}} = D_{\mathrm{v\text{-}a}}/\phi_{\mathrm{ha}}. \tag{31}$$

An independent confirmation of the diffusion enhancement of Eq. (31) is achieved through a one-dimensional heat transfer
analysis of the layered microstructure (Hansen and Foslien (2015)). This approach is attractive as there is no need to introduce the notion of intrinsic time or an intrinsic diffusion velocity. To begin, note that the energy flux of the mixture is identical to the energy flux of the individual constituents, i.e.,

$$q_{\mathrm{s}} = q_{\mathrm{i}} = q_{\mathrm{ha}}. \tag{32}$$

Whereas the constituent energy fluxes are equal, the temperature gradients are not. Recognizing this, the energy flux of the
ice is attributed to heat conduction leading to

$$q_{\mathrm{i}} = -k_{\mathrm{i}}\left(\frac{\partial\theta}{\partial x}\right)_{\mathrm{i}}. \tag{33}$$

The energy flux of the humid air is attributed to conduction of the humid air and the mass flux of water vapor. Following Bird and Lightfoot (1960) one can write

$$q_{\mathrm{ha}} = -\left(k_{\mathrm{ha}} + u_{\mathrm{sg}}D_{\mathrm{v\text{-}a}}\frac{\mathrm{d}\gamma_{\mathrm{v}}}{\mathrm{d}\theta}\right)\left(\frac{\partial\theta}{\partial x}\right)_{\mathrm{ha}}. \tag{34}$$



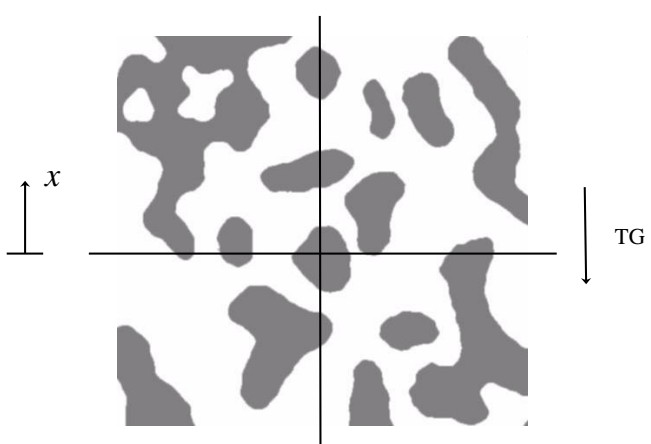

**Figure 5.** Surface section for snow showing horizontal and vertical test lines for stereology.

Combining Eqs. (30) and (32-34) leads to

$$q_s = -\left( \frac{\left( k_{ha} + u_{sg} D_{v\text{-}a} \frac{d\gamma_v}{d\theta} \right)}{\phi_i \left( \frac{k_{ha} + u_{sg} D_{v\text{-}a} \frac{d\gamma_v}{d\theta}}{k_i} \right) + \phi_{ha}} \right) \left( \frac{\partial \theta}{\partial x} \right). \tag{35}$$

An order of magnitude analysis reveals the first term in the denominator of Eq. (35) may be neglected leading to

$$q_s = -\left( \frac{k_{ha}}{\phi_{ha}} + u_{sg} \frac{D_{v\text{–}a}}{\phi_{ha}} \left( \frac{d\gamma_v}{d\theta} \right) \right) \left( \frac{\partial \theta}{\partial x} \right). \tag{36}$$

Comparing Eq. (36) with the one-dimensional form of Eq. (13) reveals the expression for the diffusion coefficient given in
Eq. (31), providing an independent confirmation of the diffusion coefficient for a layered ice/humid air microstructure.

## 3.2   1-D vapor diffusion path in snow

Consider a typical surface section of snow from a vertical slice through a RVE, Figure 5. Relative to the surface section,
horizontal and vertical test lines are identified for a stereology analysis. A strong negative temperature gradient is assumed in
the $x$ (vertical) direction.

To develop a vapor diffusion model for snow, two important results are noted. First, the vapor velocity vectors are assumed
closely aligned with the $x$ direction. An excellent visualization of this 1-D velocity field is seen in Figure 3 of Pinzer et al.
(2012), showing ice voxel displacement fields from particle image velocimetry. The water vapor mass flux is inferred in the
opposite direction of the displacement field. The figure indicates that vapor diffusion velocity is largely *a one-dimensional*
*vector field occurring within a three-dimensional ice network*. This experimental observation suggests a strong presence of the
hand-to-hand diffusion mechanism described by Yosida (1955) and is perhaps the single most important aspect of the vapor
diffusion analysis developed herein.





Second, consider the horizontal and vertical test lines through the surface section of Figure 5. $L_{\text{ha}}$ is identified as the distance a line segment is in the humid air phase while $L_{\text{T}}$ is the total test line length. Then for both vertical and horizontal directions, one can write

$$\phi_{\text{ha}} = (L_{\text{ha}}/L_{\text{T}}). \tag{37}$$

Hence, horizontal test lines revealing ice blockage and vertical test lines showing shortened diffusion paths occur with the same frequency, a harbinger of the relative influence of these diffusion mechanisms.

The mass flux in the humid air at the microscale is given by Eq. (17). Volume averaging Eq. (17) over the entire RVE (ice and humid air) leads directly to

$$\phi_{\text{ha}}\gamma_{\text{v}}v_{\text{v}} = -\phi_{\text{ha}}D_{\text{v-a}} \left(\frac{d\gamma_{\text{v}}^{\text{sat}}}{d\theta_{\text{ha}}}\right)\left(\frac{\partial\theta}{\partial x}\right)_{\text{ha}}. \tag{38}$$

Now consider an arbitrary point in the horizontal $(y-z)$ plane of the RVE. As that point is swept along the $x$ direction, diffusing water vapor molecules will encounter ice grains as shown in Figure 5. Under the assumption of a 1-D diffusion flow field, the diffusion paths are shortened by $\phi_{\text{ha}}$ in a manner consistent with the surface section of Figure 5. Hence, dividing Eq. (38) by $\phi_{\text{ha}}$ and introducing the intrinsic diffusion velocity gives

$$j_{\text{s}} = \phi_{\text{ha}}\gamma_{\text{v}}v_{\text{v}}^* = -D_{\text{v-a}} \left(\frac{d\gamma_{\text{v}}^{\text{sat}}}{d\theta_{\text{ha}}}\right)\left(\frac{\partial\theta}{\partial x}\right)_{\text{ha}}. \tag{39}$$

Comparing the above equation with the 1-D form of Eq. (11) gives

$$D_{\text{s}}\left(\frac{\partial\theta}{\partial x}\right) = D_{\text{v-a}}\left(\frac{\partial\theta}{\partial x}\right)_{\text{ha}}, \tag{40}$$

The implication of Eq. (40) is profound. Because the thermal conductivity of ice is greater than that of humid air, the humid air temperature gradient is greater than or equal to the temperature gradient of the snow mixture. This implies $D_{\text{s}} \geq D_{\text{v-a}}$, indicating diffusion in snow is enhanced compared to diffusion of water vapor in air only.

The underlying assumptions leading to Eq. (40) are that water vapor transport in snow is diffusion limited and the diffusion path is one-dimensional. In this case, the influence of ice blockage is countered precisely by the intrinsic diffusion velocity resulting from a shortened diffusion path. We again refer to the results of Pinzer et al. (2012), suggesting a 1-D diffusion path for water vapor in snow is a reasonable assumption.

Noting the temperature gradient constituent relation of Eq. (16), Eq. (40) may be expanded as

$$D_{\text{s}}\left(\phi_{\text{i}}\left(\frac{\partial\theta}{\partial x}\right)_{\text{i}} + \phi_{\text{ha}}\left(\frac{\partial\theta}{\partial x}\right)_{\text{ha}}\right) = D_{\text{v-a}}\left(\frac{\partial\theta}{\partial x}\right)_{\text{ha}}. \tag{41}$$

To proceed further, a relationship between the ice and humid air constituent temperature gradients is required. Acquiring this information would require thermal analyses of multiple RVE's as was done by Calonne et al. (2011) and Riche and Schneebeli (2013). In the absence of this level of numerical modeling, a reasonable estimate of the relationship between constituent temperature gradients may be obtained from a thermal conductivity analysis.




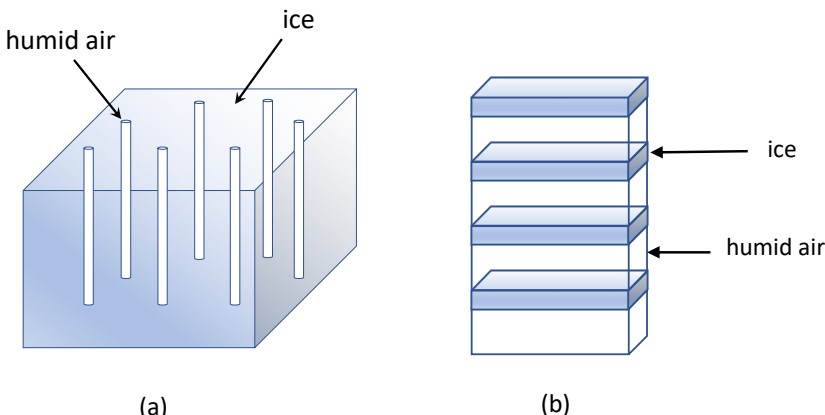

(a)  (b)

**Figure 6.** (a) Pore microstructure (b) Layered microstructure

Foslien (1994) developed a thermal conductivity model for snow based on a linear combination of the pore and layered microstructures shown in Figure 6 combined with arguments from stereology.

The thermal conductivity relation of Foslien (1994) assumes the form

$$k_{\text{s}} = \phi_{\text{i}} k_{\text{pore}} + \phi_{\text{ha}} k_{\text{lay}}. \tag{42}$$

Relations for $k_{\text{pore}}$ and $k_{\text{lay}}$ are developed in Hansen and Foslien (2015) leading to

$$k_{\text{s}} = k_{\text{ha}} \left( 1 + \phi_{\text{i}} \phi_{\text{ha}} \right) + k_{\text{i}} \phi_{\text{i}}^2. \tag{43}$$

Figure 7 provides the predictions of Eq. (43) for a temperature of 253 K against the curve fit of Riche and Schneebeli (2013). The correlation of the analytical model is excellent as the model virtually tracks the numerical results.

Following the form of Eq. (42), the constituent temperature gradients are assumed to be weighted identically to the influence of the pore and layered microstructures as

$$\left( \frac{\partial \theta}{\partial x} \right)_{\text{i}} = \phi_{\text{i}} \underbrace{\left( \frac{\partial \theta}{\partial x} \right)_{\text{i}}}_{\text{pore}} + \phi_{\text{ha}} \underbrace{\left( \frac{\partial \theta}{\partial x} \right)_{\text{i}}}_{\text{layer}}. \tag{44}$$

Note that, for the pore microstructure

$$\left( \frac{\partial \theta}{\partial x} \right)_{\text{i}} = \left( \frac{\partial \theta}{\partial x} \right)_{\text{ha}}, \tag{45}$$

whereas for the layered microstructure

$$\left( \frac{\partial \theta}{\partial x} \right)_{\text{i}} \approx 0. \tag{46}$$


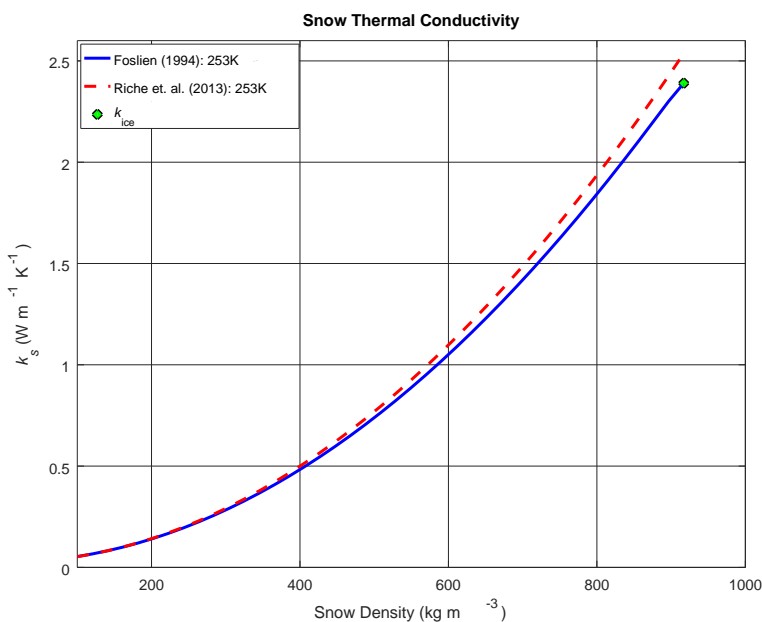

**Figure 7.** Thermal conductivity analytical prediction of Foslien (1994) versus finite element predictions of Riche and Schneebeli (2013).

Combining Eq. (41) with Eqs. (44-46) leads to an analytical expression for the diffusion coefficient, sans tortuosity effects, given by

$$D_{\text{s}} = \frac{D_{\text{v-a}}}{\phi_{\text{ha}} + \phi_i^2}. \tag{47}$$

### 3.3  Vapor diffusion including tortuosity effects

5   To account for the influence of tortuosity, the work of Calonne et al. (2011) for computing thermal conductivity is introduced. Heat transfer at the microscale is defined by the boundary value problem over the RVE as

$$
\begin{aligned}
\boldsymbol{\nabla} \cdot (k_{\text{i}} \boldsymbol{\nabla} \boldsymbol{t}_{\text{i}} + \boldsymbol{I}) &= \boldsymbol{0} && \text{in } \Omega_{\text{i}} \\
\boldsymbol{\nabla} \cdot (k_{\text{ha}} \boldsymbol{\nabla} \boldsymbol{t}_{\text{ha}} + \boldsymbol{I}) &= \boldsymbol{0} && \text{in } \Omega_{\text{ha}} \\
\boldsymbol{t}_{\text{i}} - \boldsymbol{t}_{\text{ha}} &= \boldsymbol{0} && \text{on } \Gamma \\
k_{\text{i}} (\boldsymbol{\nabla} \boldsymbol{t}_{\text{i}} + \boldsymbol{I}) - k_{\text{ha}} (\boldsymbol{\nabla} \boldsymbol{t}_{\text{ha}} + \boldsymbol{I}) \cdot \boldsymbol{n} &= \boldsymbol{0} && \text{on } \Gamma \\
\frac{1}{V} \int_{V} (\boldsymbol{t}_{\text{i}} + \boldsymbol{t}_{\text{ha}}) \, dV &= \boldsymbol{0},
\end{aligned}
\tag{48}
$$



where $t_\mathrm{i}$ and $t_\mathrm{ha}$ are vectors describing the microscale fluctuations in the temperature gradients of the ice and humid air, respectively, and $I$ is the identity tensor.

The tensorial form of the thermal conductivity for snow is defined in Calonne et al. (2014) as

$$k_\mathrm{s} = \frac{1}{V} \left( \int\limits_{V\mathrm{ha}} k_\mathrm{ha} \left( \boldsymbol{\nabla} t_\mathrm{ha} + I \right) dV + \int\limits_{V\mathrm{i}} k_\mathrm{i} \left( \boldsymbol{\nabla} t_\mathrm{i} + I \right) dV \right). \tag{49}$$

Because the humid air is saturated, the microscale diffusion problem is driven by the known microscale temperature gradient. Hence, the solution to the diffusion problem must produce the same microscale temperature gradient field as the thermal problem.

The diffusion solution is obtained by using the solution for the thermal problem with the following substitutions

$$\begin{aligned} k_\mathrm{ha} &\to D_\mathrm{v\text{-}a}, \\ k_\mathrm{i} &\to D_\mathrm{v\text{-}a} \left( \frac{k_\mathrm{i}}{k_\mathrm{ha}} \right). \end{aligned} \tag{50}$$

The above substitutions represent a major departure from Calonne et al. (2014) who assumed

$$\begin{aligned} k_\mathrm{ha} &\to D_\mathrm{v\text{-}a}, \\ k_\mathrm{i} &\to 0. \end{aligned} \tag{51}$$

Justifcation for the substitutions given by the relations of (50) is as follows. First, dividing the relations of (50) gives

$$\left( \frac{k_\mathrm{i}}{k_\mathrm{ha}} \right) \to \left( \frac{k_\mathrm{i}}{k_\mathrm{ha}} \right). \tag{52}$$

Preserving the ratio of $\left( \frac{k_i}{k_{ha}} \right)$ is a necessity to preserve the local temperature gradient field obtained from the heat transfer

analysis.

An excellent physical interpretation of the latter substitution of (50) is that $D_\mathrm{v\text{-}a} \left( \frac{k_i}{k_{ha}} \right)$ may be thought of as a fictitious diffusion velocity for the ice phase. Hence, one can envision the entire RVE as a diffusing medium for water vapor where the diffusion velocity in the ice phase is nearly 100 times larger than the diffusion velocity through the humid air.

Again, the work of Pinzer et al. (2012) is relied on, showing a largely one-dimensional (not purely) diffusion velocity field

for water vapor transport in snow. Then to a *first approximation*, diffusion paths are shortened according to $L_\mathrm{ha} = \phi_\mathrm{ha} L_\mathrm{T}$ due to the ice phase.

Recognizing the influence of intrinsic time on water vapor diffusion, the tensorial form of the diffusion coefficient for snow is obtained functionally from

$$\boldsymbol{D}_\mathrm{s} = \frac{1}{V_\mathrm{ha}} \int\limits_{V\mathrm{ha}} D_\mathrm{v\text{-}a} \left( \boldsymbol{\nabla} t_\mathrm{ha} + I \right) dV, \tag{53}$$

where $t_\mathrm{ha}$ represents the temperature gradient fluctuations of the humid air as defined by the original thermal problem of Eq. (46). Note that the averaging process is done over the humid air volume as opposed to the total volume as the effects of ice blockage and the shortened diffusion paths from hand-to-hand vapor transport cancel each other out.





Given a collection of snow RVE's and a solution of the thermal boundary value problem for the local temperature gradient field, Eq. (53) is perhaps the best means of determining accurate values for the water vapor diffusion coefficient in snow. However, in the absence of this numerical data, a diffusion model including the effects of tortuosity is obtained by again referencing the thermal conductivity solution of Hansen and Foslien (2015) given by Eq. (43). Using this equation and the

diffusion analogy of Eq. (48) gives

$$D_s = D_{v-a}\left(1 + \phi_i \phi_{ha}\right),\tag{54}$$

where the artificial diffusion term for the ice phase is neglected. The sole purpose of this artificial diffusion term is to produce a local temperature gradient field consistent with the thermal boundary value problem.

### 3.4   Discussion

The three mathematical models for the diffusion coefficient of water vapor in snow developed herein are summarized below.

1. Upper bound

$$D_s = D_{v\text{-}a}/\phi_{ha}.\tag{55}$$

2. 1-D diffusion path

$$D_s = \frac{D_{v\text{-}a}}{\phi_{ha} + \phi_i^2}.\tag{56}$$

3. Diffusion with tortuosity

$$D_s = D_{v-a}\left(1 + \phi_i \phi_{ha}\right),\tag{57}$$

Figure 8 shows the 3 diffusion coefficient curves predicted by the above models. Several interesting features of these curves are noteworthy. First, the upper bound produces a modest ceiling of 2 on the normalized diffusion coefficient for ice volume fractions from $0 \rightarrow 0.5$ (densities up to $460\,\mathrm{kg\,m^{-3}}$). This ceiling is significantly lower than the experimental findings of Yosida

(1955) and the theoretical results of Colbeck (1993). At an ice volume fraction of 0.3, the upper bound is much tighter, showing a maximum normalized diffusion coefficient of 1.43.

The curve of the 1-D vapor transport model is significant, as the relative influence of ice blockage and the intrinsic diffusion velocity from shortened diffusion paths effectively cancel each other, leaving only the elevated temperature gradient in the humid air as a mechanism enhancing diffusion. At low densities, the small ice volume fractions do not produce an elevated humid

air temperature gradient, resulting in a normalized diffusion coefficient near 1. At very high densities, the highly connected ice phase produces temperature gradients in ice that are comparable to that of the humid air, resulting in near equal constituent temperature gradients with the macroscale snow temperature gradient. In essence, heat transfer in snow begins to look like the pore microstructure of Figure 6(a) leading to a normalized diffusion coefficient of 1.

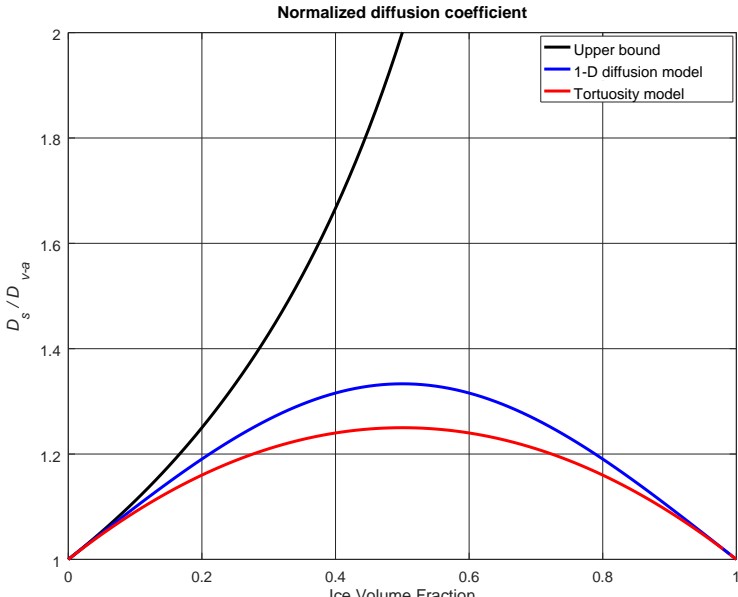

**Figure 8.** Predicted diffusion coefficient curves including (a) upper bound, (b) 1-D diffusion velocity field, and (c) diffusion including tortuosity

.

The diffusion curve based on a 1-D velocity field and the curve incorporating tortuosity effects are also appealing in that the curves are nearly identical for low density snow where tortuosity should not be a factor. Furthermore, like the 1-D vapor transport model, the tortuosity curve approaches unity as $\phi_i \rightarrow 0$. Interestingly, the tortuosity curve also approaches unity at very high ice volume fractions. An explanation for this effect is that at high densities, diffusing water vapor is unable to follow a continuous path and may well be isolated within air pockets in the ice. As a result, diffusion must follow a more one-dimensional path. Finally, note the appealing result that the diffusion curve including tortuosity is below the 1-D diffusion curve over the entire range of ice volume fractions from $0 \rightarrow 1$.

The curves of Figure 8 suggest a reasonable estimate of the normalized diffusion coefficient is that it lies in a range from $1 - 1.3$. Given the historical range of predictions for this parameter, the predicted range is quite narrow and suggests a mild enhancement for water vapor diffusing through snow compared to diffusion in humid air alone.

At this point, it is interesting to revisit prior studies to place them in the context of the analytical models presented above. Pinzer et al. (2012) estimated the mass flux in snow using a heat transfer finite element simulation of a RVE under a prescribed

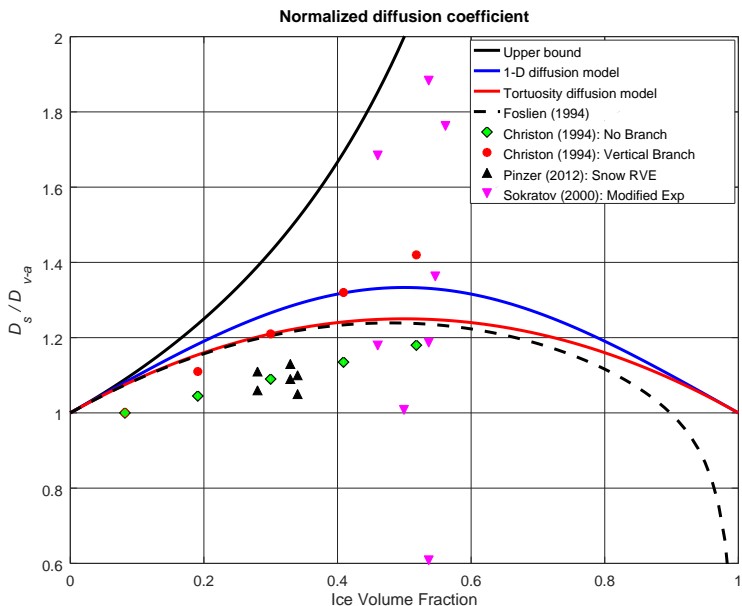

**Figure 9.** Predicted diffusion coefficient curves including numerical predictions and experimental data.

temperature gradient. The mass flux is determined directly from the local temperature gradient field, an approach consistent with the analytical work developed herein.

Pinzer et al. (2012) assumed the local influence of latent heat can be neglected as the heat is immediately conducted away by the highly conductive ice. They also assumed the mass flux is limited only by the diffusion velocity of the water vapor,

i.e., attachment kinetics of condensing water molecules is not a factor. Support for this assumption can be found in Foslien (1994). Using an expression from Hobbs (1974) for vapor condensation in terms of the vapor density difference between the pore vapor density and the saturated vapor density, Foslien showed the time for the vapor density difference to become 0.1% of the the initial vapor density difference is on the order of $10^{-3}s$. Given diffusion times of interest on the order of hours and days, this result suggests the assumption of a diffusion limited process is reasonable.

Pinzer et al. (2012) computed the diffusion coefficient for snow by averaging the humid air mass flux over several slices of the RVE perpendicular to the temperature gradient. This approach is equivalent to the averaging process of Eq. (53). Figure 9 shows the finite element predictions of Pinzer et al. (2012) for the vapor diffusion coefficient. No adjustment of their data is necessary as Eq. (53) addresses all diffusion mechanisms considered in the analytical models.

Christon et al. (1994) performed some of the first finite element modeling of heat and mass transfer for ice/humid air lattices

at the microscale in an effort to predict macroscale thermophysical properties for snow. Of interest here is the calculation for





the diffusion coefficient given by

$$D_{\mathrm{s}} = \frac{\dot{m}\Delta x}{A_{\mathrm{Tot}}\Delta\gamma_{\mathrm{v}}}, \tag{58}$$

where $\dot{m}$ is the mass flow rate and $A_{\mathrm{Tot}}$ is the total cross-sectional area of the unit cell.

Equation (58) accounts for the influence of ice blockage impeding diffusion through the use of the total area, $A_{\mathrm{Tot}}$, but does

not account for diffusion enhancement due to shortened diffusion paths.

Figure 9 shows the diffusion coefficient predictions of Christon et al. (1994) for two different ice/humid air microstructures. Any correction for the shortened diffusion paths is not readily possible as Christon's ice/humid air microstructures contain regions where the humid air is unobstructed, as in the pore microstructure, and therefore exhibit no diffusion enhancement due to shortened diffusion paths. In other areas of the microstructure, there are regions where the ice clearly shortens the diffusion

path. At best, one can say that a correction of Christon's data for shortened diffusion paths, manifest as an elevated intrinsic diffusion velocity, would produce an enhancement multiplier of $\alpha$ where $1 < \alpha < (1/\phi_{\mathrm{ha}})$. This correction of Christon's results would not significantly alter the conclusions of their data shown in Figure 9.

The agreement of the results of Christon et al. (1994) with the diffusion models presented is not surprising, despite the simpler ice/humid air microstructures they utilized. Note that the mass flux for a RVE is driven by the volume averaged

temperature gradient of the humid air which is dictated by a linear boundary value problem. Volume averaging of microscale quantities in a linear problem is a very forgiving process in terms of accuracy, as integration is a smoothing process. This same phenomenon has been observed in composite materials where elastic constants obtained by volume averaging local stress/strain fields have shown little variation between random fiber distributions in RVE's and periodically structured unit cells for the same material (Brockenbrough et al. (1991)).

Perhaps the most extensive numerical study of mass transport is found in Calonne et al. (2014). They performed finite element analyses for 35 RVE's of snow spanning a density range from $100 - 500\,\mathrm{kg\,m}^{-3}$. Unfortunately, it is not possible to reconcile their results with the analytical models presented, as they solved a fundamentally different boundary value problem for mass transfer driven by the relation substitutions shown in (51) where the thermal conductivity of ice was set to zero. In contrast, in the analytical models, the influence of $k_{\mathrm{i}}$ is retained to obtain the local temperature gradient field. The influence of

$k_{\mathrm{i}}$ is then removed in the calculation of the diffusion coefficient.

Sokratov and Maeno (2000) conducted a series of experiments to determine the diffusion coefficient of snow. Their approach similarly followed the path of Christon et al. (1994) in that they accounted for the effects of ice blockage impeding diffusion but did not account for the shortened diffusion paths resulting from hand to hand vapor transport. Since the experiments were done on snow, the present work shows the diffusion enhancement resulting from shortened pathways, manifest as intrinsic

time, is obtained by dividing the results of Sokratov and Maeno (2000) by $\phi_{\mathrm{ha}}$. The modified data reflecting this correction is shown in Figure 9. Although the scatter in the data is significant, the results are generally in agreement with values predicted by the diffusion models presented.

The theoretical work of Colbeck (1993) has drawn significant attention as it represents an attempt to model the hand-to-hand diffusion mechanism of water vapor transport between ice grains. Colbeck suggests that the blocking effect of ice is "of little





consequence" and hence is neglected. However, the analysis shown here indicates the influence of ice blocking diffusion paths is of equal importance to diffusion enhancement of shortened paths caused by particle-to-particle water vapor transport. If one adjusts the model of Colbeck (1993) by scaling his model by $\phi_{\mathrm{ha}}$, the ice blocking effect, the predicted diffusion coefficient still dramatically exceeds the upper bound noted in Figures 8 and 9.

Foslien's (1994) model of the diffusion coefficient tracks at or just below the tortuosity diffusion model of Figure 9 over the density range from $0 - 800\,\mathrm{kg\,m^{-3}}$ before crashing toward zero as the ice volume fraction approaches one. The model is intriguing as it is arrived at through an entirely different approach from the microscale volume averaging used to develop the analytical models in this work.

## 4   Conclusions

I have attempted to develop analytical models for the diffusion coefficient of water vapor in snow that account for all major diffusion mechanisms caused by the introduction of an ice phase in humid air. The models suggest snow enhances diffusion at all densities but that enhancement is rather minimal, showing a typical normalized diffusion coefficient of snow with respect to humid air alone in the range $1 \rightarrow 1.3$.

The chief difficulty for any study of the diffusion coefficient is to properly account for all diffusion mechanisms that are

introduced by the presence of the ice phase. Arguments from stereology, combined with experimental observation, suggest that the influence of ice blocking diffusion paths is canceled out by the shortened diffusion paths from hand to hand vapor transport.

Accounting for the influence of tortuosity is not immediately obvious. Given the numerical firepower of today and the ability to produce accurate snow RVE's for finite element analysis, it is a straight forward matter to compute tortuosity of a porous solid such as rock and air. However, in the case of snow, the condensation and sublimation of water vapor mitigates

tortuosity, as the hand-to-hand particle interaction effectively straightens out diffusion paths. It is this phenomenon that leads to the experimental observation that diffusion of water vapor in snow resembles a one-dimension vapor velocity field in a three-dimensional ice/humid air microstructure.

In closing, the wild complexities involving the study of the water vapor diffusion coefficient in snow may be distilled down to a few equations. To begin, the mass flux of water vapor diffusing across a surface in snow is given by

$$\boldsymbol{j}_{\mathrm{s}} = \gamma_{\mathrm{v}} \boldsymbol{v}_{\mathrm{v}} = \frac{1}{V_{\mathrm{ha}}} \int\limits_{V_{\mathrm{ha}}} \gamma_{\mathrm{v}}\left(\boldsymbol{\xi}\right) \boldsymbol{v}_{\mathrm{v}}\left(\boldsymbol{\xi}\right) dV_{\mathrm{ha}}. \tag{59}$$

Importantly, Eq. (59) represents the mass flux per unit area of *snow* and not just the flux across the humid air phase as the ice blockage impeding diffusion is exactly countered by the shortened diffusion paths of water molecules. The expression $\gamma_{\mathrm{v}} \boldsymbol{v}_{\mathrm{v}}$ is based on the true macroscale diffusion velocity in the humid air constituent. The intrinsic diffusion velocity does not appear as its sole purpose was to provide a mathematical means of accounting for altered lengths of diffusion paths through a RVE.





Second, given an RVE for snow with an imposed 1-D macroscale temperature gradient of unity, the diffusion coefficient is defined by

$$D_{\mathrm{s}} = \frac{1}{V_{\mathrm{ha}}} \int_{V_{\mathrm{ha}}} D_{\text{v-a}} \left( \frac{\partial \theta(\boldsymbol{\xi})}{\partial \xi_1} \right) dV$$

$$= D_{\text{v-a}} \left( \frac{\partial \theta}{\partial x} \right)_{\mathrm{ha}}, \tag{60}$$

5 where the volume averaging takes place over the humid air, only.

When all diffusion mechanisms are properly accounted for, the experimental and numerical data, as well as theoretical models, presented in Figure 9 produce a consistent result, showing snow has a slightly enhanced diffusion coefficient over all densities.

There is ample room for improvement of the theoretical models presented for the diffusion coefficient. An area of interest
10 is the development of models for an anisotropic diffusion coefficient tensor. Calonne et al. (2014) show that there is definitely anisotropic diffusion behavior, particularly in the case of depth hoar. Another area of importance is seeking relevant microstructural parameters beyond constituent volume fractions to improve parameterized models of diffusion, as was done by Löwe et al. (2013) for thermal conductivity.





**Appendix A: List of Symbols**

| | |
|---|---|
| $dS$ | Differential surface area ($\mathrm{m}^2$) |
| $\boldsymbol{I}$ | Identity tensor |
| $V$ | Volume ($\mathrm{m}^3$) |
| $t$ | Time ($\mathrm{s}$) |
| $C_\alpha^V$ | Specific heat of constituent $\alpha$ at constant volume ($\mathrm{J\,kg^{-1}\,K^{-1}}$) |
| $D_{v-a}$ | Diffusion coefficient of water vapor in air ($\mathrm{m^2\,s^{-1}}$) |
| $D_\mathrm{s}$ | Diffusion coefficient of snow ($\mathrm{m^2\,s^{-1}}$) |
| $\boldsymbol{j}_v$ | Diffussive flux of water vapor (microscale)($\mathrm{kg\,m^{-2}\,s^{-1}}$) |
| $\boldsymbol{j}_s$ | Diffussive flux of water vapor (macroscale) ($\mathrm{kg\,m^{-2}\,s^{-1}}$) |
| $k_\alpha$ | Thermal conductivity of constituent $\alpha$ ($\mathrm{W\,m^{-1}\,K^{-1}}$) |
| $k_\mathrm{s}$ | Thermal conductivity of snow ($\mathrm{W\,m^{-1}\,K^{-1}}$) |
| $\boldsymbol{q}_\mathrm{s}$ | Energy flux of snow ($\mathrm{W\,m^{-2}}$) |
| $\boldsymbol{q}^\mathrm{c}$ | Energy flux due to conduction ($\mathrm{W\,m^{-2}}$) |
| $\boldsymbol{q}^\mathrm{d}$ | Energy flux due to interdiffusion of species ($\mathrm{W\,m^{-2}}$) |
| $\boldsymbol{t}_\mathrm{i}$ | Vector characterizing temperature fluctuation in the ice phase |
| $\boldsymbol{t}_\mathrm{ha}$ | Vector characterizing temperature fluctuation in the humid air phase |
| $t^*$ | Intrinsic time ($\mathrm{s}$) |
| $u_\mathrm{sg}$ | Latent heat of sublimation of ice ($\mathrm{J\,kg^{-1}}$) |
| $\boldsymbol{v}_\mathrm{v}$ | Velocity of water vapor component ($\mathrm{m\,s^{-1}}$) |
| $\boldsymbol{v}_\mathrm{v}^*$ | Intrinsic velocity of water vapor component ($\mathrm{m\,s^{-1}}$) |
| $\gamma_\alpha$ | True density of constituent $\alpha$ ($\mathrm{kg\,m^{-3}}$) |
| $\phi_\alpha$ | Volume fraction of constituent $\alpha$ |
| $\theta$ | Temperature ($\mathrm{K}$) |
| $\rho_\alpha$ | Dispersed density of constituent $\alpha$ ($\mathrm{kg\,m^{-3}}$) |
| $\boldsymbol{\xi}$ | Spatial coordinates at the microscale |
| $\theta_\alpha$ | Temperature of constituent $\alpha$ ($\mathrm{K}$) |
| $\nabla$ | Gradient operator |

**Subscripts**

| | |
|---|---|
| $\alpha$ | Arbitrary constituent |
| ha | Humid air constituent |
| i | Ice constituent |
| s | Snow |
| v | Water vapor component |
| $\xi$ | Indicates variable applies at the microscale |





*Author contributions.* The author is soley responsible for the research and writing of this manuscript.

*Competing interests.* The author has no conflict of interest.



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
