# Peer review of "Revisiting the vapor diffusion coefficient in dry snow"

_The Cryosphere, 2019_

## Short Comment (SC1) · 16 Jul 2019

Hello A. C. Hansen,

This paper is fascinating. Specifically the "Hand to Hand" idea and introducing the the ice phase into the direction of vapour travel within the porous space. And short pathways during metamorphic process. Conceptually it would make sense that local temperature gradients would be stronger between ice grains and it's bottom grain surface, almost "creating" a pathway of lesser resistance given sublimation, than that of just humid air.

I forecast avalanches professionally in British Columbia, and in a very rough way, think about these concepts and weather impact on stability of slab layers.

[Figure]

Would you consider doing a video abstract on this article, so that I could share this with my colleagues in a slightly more digestible way. It would be great to see this reach a wider audience for discussion within the avalanche community. This is the kind of information that leads to and supports building rules of thumb and indicators of accurate forecasting.

Lee

---

## Author Comment (AC1) · 17 Jul 2019

Hello Lee,

Thanks very much for your note and your thoughts on: "Revisiting the vapor diffusion coefficient in dry snow." I also find the subject to be fascinating and, at times, a mind bending exercise. It is a challenge to account for all the competing mechanisms that both enhance and impede the diffusion process.

Sincerely,

Andrew Hansen
* * *

---

## Referee Comment (RC1) · Anonymous Referee #1 · 29 Oct 2019

**1   General comments**

This paper addresses a long standing and vividly debated topic in the snow science community and makes a contribution by unifying the definition of effective diffusion. The paper converts the wide variety of predicted and measured effective diffusive coefficient $D_s$ to reasonable range.  The paper starts from a mixture theory representation of transport of mass and energy in snow and defines an upper bound for the effective vapor diffusion coefficient as a function the density of snow. By introducing an objective definition of vapor transport one can compare several previous experimental studies on effective vapor transport. The paper shows that all measurements are well below the upper bound. In addition it formulates two models based on diffusion with and without

tortuosity.

I agree with the author that effective vapor diffusion is a mind boggling topic and can become confusing from time to time. The author attempts to make a first step in clearing the clutter by starting from the first principles of mixture theory. He identifies 4 ways the diffusion path of water can be altered with respect to pure vapor diffusion without an ice matrix interfering. 1) Blockage of diffusion paths by ice, 2) Shortened diffusion paths due to the phase transition at the interface. 3) Enhanced local diffusion by difference in heat conduction in both phases. 4) Tortuosity, lengthening diffusion paths. The author states that any model for the effective diffusion coefficient these 4 principles should be accounted for.

The author introduces an artificial velocity of the vapor molecules that is calculated by the path length that it travels divided by the time spend in the vapor phase only, effectively deleting the time it spends in the ice phase. Hereby the definition of the effective diffusion coefficient is enhanced. Based on this concept, the author defines an upper bound for $D_s$, which is a density correction based on this ice volume fraction.

The first model, which is a superposition of a layered and a tubular microstructure, leads to a slightly different density correction. The last model that should include tortuosity and is based on the method developed by Calonne et al. (2014). They calculate the effective diffusion by changing transport coefficients from the heat conductivity problem in terms of the transport of the vapor flux coefficients. Here the transport in the ice phase was set to 0. In this work the ice transport coefficient is set to 100 times that of vapor, to implement the idea of shortened pathways or intrinsic/artificial velocity.

My main concern is on the introduction of the intrinsic time or intrinsic/artificial velocity. It is introduced to delete the time the vapor molecules spend in the ice phase, yielding an effective higher velocity to the vapor molecules, since the distance is that it apparently travelled through the ice phase is still counted. If this approach of apparent mass transfer is taken, the apparent movement of the ice matrix in opposite direction of the

flux should be accounted for as well. The latter is important, because without it the mass balance of your snow pack wouldn't be complete. The intrinsic time is confusing and as concluded in the end of the paper is not really necessary. However, it is still the basis to compute the effective diffusion coefficient, which raises the question if the meaning of this definition is physical or not.

The main crux is to assess whether the hand-to-hand vapor transport is physically related to mass flux. Since the water molecules diffusivity in ice is orders of magnitudes slower giving it a large of infinite average velocity is not physical. Mechanisms to which actual transport is being enhanced is if the local vapor flux in the pore space is on average more enhanced, than the reductions that are expected. It is known that tortuosity $\tau$ is leading to an upper bound for $D_s^* = (1 - \phi_i)D_{v-a}/\tau$ (Pismen, 1974). Note that this is only a reduction of $D_{v-a}$. A heterogeneous distribution of temperature gradients in porous media will lead to actual locally enhanced vapor flux. Bounds for these fields could be found/developed based on e.g. Torquato (2002), but as far as I know, cannot exceed a density correction factor, as the author suggests for his upper bound.

My interpretation of the given models is that they use the linearization between the vapor concentration gradient and the temperature gradient at the microscale and find density corrections for the latter given 1) a layered, 2) a mix between tubular and layered microstructures. The physical interpretation of the last model is not clear to me, especially how tortuosity is included. If the models are restated in the context of enhanced temperature gradients they might be more physical/useful.

In conclusion, the debate around the effective diffusion coefficient is in need for a uniform and clear definition. The approach which is chosen in this paper, is interesting, but also highly confusing. In my opinion the effective diffusion coefficient is a linear response of water vapor and ice transport to a thermal driving force. Since the phase transitions that take place at the microscale serve as a temporal storage of vapor, i.e. ice, it should in principle reduce the effective transport, and therefore reduce the effective diffusion coefficient. Given that it is the authors choice to define it in a different manner, I will respond with some specific comments below.

**2 Specific comments**

**2.1 Section 1: History**

- General: Introduction is quit sparse regarding the explanation of the different mechanisms for enhanced vapor diffusion. For understanding the introduction can be more detailed. example:

- p.2. l.6: More explanation of this experiment of Sommerfeld et al. (1987) would clarify how this was measured, which assumptions are made etc...same for the studies Pinzer et al. (2012) and Calonne et al. (2014)

- p.2 l.32: What is physical difference between blockage of diffusion paths and tortuosity?

- General: From the history or the introduction, it should become clear to the reader which studies have been performed and where the discrepancies lie. To me it is not clear without reading the actual referenced papers what has been measured, modeled or simulated. Suggestion: start with stating that differences in studies are mainly based on the assumptions of contributing mechanisms and the lack of a consistent definition, and depart from there by describing which assumptions they made.

**2.2 Section 2: Mass transfer in snow**

- p.5. l.8: If isotropy can be relaxed please discuss it somewhere at a later stage.

[Figure]

- p.6. l.11: The introduction of apparent time is confusing to me. If we were to couple measured diffusion to mass transport, we would need to add the residence time spend in the crystal to the time travelled in humid air? If you make the connection to residence time, one could 'track' single molecules.

- p.7. l.15: With eq.11 the definition of $D_s$ is given, which is basically the effective linear response to the physical driving force. Maybe consider this as a natural bridge to describe what it is effected by given the physics at the microscale.

- p.7. l.20: In this paragraph we suppose to be convinced that it is useful to frame $D_s$ in connection to the balance equations. We know that they are dependent on $D_s$ but a quick calculation of the relative importance of the second term is less than 5 ‰.

- p.8. l.9: Despite the title, there is no formal definition of $D_s = \ldots$. One could use eq.(11) but then it means that is connected to the flux of snow, which can only be understood in terms of mass flux.

- p.8. Fig.3: This figure is confusing, suggesting that the bonds are colder than the grains, by the colorbar. The RVE is also on the small side, usually one expects the system size to be roughly 10 times the element (grain) size.

- p.9. l.15: By taking out $d\gamma_v/d\theta$ of the integral, you make quite some assumptions, i.e. the temperature gradient and the vapor gradient are locally always reacting the same, regardless of its surroundings. In principle one should solve the Laplace equation for your vapor density, given the appropriate boundary conditions, which can include the interface dynamics of the phase transition. Stating clearly which assumptions you make is in place here.

- p.9. l.22: Suggesting linking macro to micro scale by explicitly relating $j_s$ to $\overline{j_v}$, which gives you an expression for $D_s$ that shows by what physics it is influenced.

- p.10 l.1: I believe 1) and 4) are in principle the same. From a microstructural perspective I would suggest that there are two mechanisms: 1) Tortuosity that increases the free path length of water vapor molecules. and 2) Phase transitions that, driven by temperature gradients at the ice-air interface, change the boundary conditions for the mass transfer of the vapor concentration and therefore alters the diffusion paths in a structural manner.

- p.10. l.9: My general thought is that it would not be correct to use $v^*$, since actual molecules aren't travelling through the ice phase with (in)finite speed, and if one looks at it from an apparent perspective, one has to account for the apparent movement of the ice crystals in the opposite direction that is represented by the ice interface velocity in opposite direction.

**2.3   Section 3: Models**

- p.14 l.4 & Fig 5: The argument that blockage and shortening of diffusion paths are equally frequent is based on the frequency of the ice phase, which is, for isotropic media the same. Given that the chance for shortening or blocking is given by underlying physics one still need to solve the vapor diffusion equation including the correct boundary conditions. Therefore the relative chances can still be very different.

- p.15. l.1: Is the underlying assumption that the microstructure is a superposition of a layered and pore structures?

- p.15. l.4: the naming Pore, is highly confusing. I would suggest something like tubular microstructure.

- p.15. l.7: There is only a slight difference in figure 7, but there is no information on how well this fit is doing to predict real conductivities.

- p.16. Fig. 7: What is the purpose of this figure? In the caption you write finite element predictions, but this should be a fit to finite element simulations.

- p.17. l.16: This is just not physical in terms of the [Calonne.2014] paper.

- p.18. l.6: This equation is not derived but stated? It is hard to follow this line of thought without understanding the appendix of Calonne et al. (2014)

- p.18. l.7: How is tortuosity exactly included? Tortuosity is generally described by the path that molecules travel through a porous media without altered paths due to phase transitions, which is basically governed by laminar flow in an inert media. How is this relevant or included in this model?

- p.19. Fig.8: Redundant figure.

- p.20. l.10: There are 6 values plotted, but I can't retrace them to measurements from the [Pinzer.2012] paper, a little more information could be helpful on how you did this, which figures or data you actually used.

2.4   Section 4: Conclusion

- As the author states eq.(59) should be the vapor mass flux in snow. There is no need for an introduction of shortened diffusion paths, or intrinsic velocity, because it is not physical. In the conclusion it is stated that in principle the definition of $D_s$ is the same as eq.59. This to me is confusing because you use $v^*$ in eq.(11). Maybe I'm missing the general point of the paper, but one way or the other this should be clarified.

Figures: In general the figures are not illustrative to the paper. Fig 3 is very confusing, suggesting bonds are colder than grains. My suggestion is to replace Fig 1,2,3 with a slice of a 3D image of actual snow and indicate the different processes leading to vapor diffusion (tortuosity and phase transitions at the interface)

**3  Technical comments**

- Notation: The author decides on using $\theta$ as a symbol for temperature and $\xi$ as symbol for coordinates. This might be the convention in mixture theory, but for readability and adaptation to the snow community I would advise on $T$ and $x$ respectively.

- Use of the word 'may' in e.g. p.7.13 sounds to me as if the followed expression is lucrative in the sense that we may also use something else. The word 'can' in this context would be more appropriate, however, it is a matter of taste.

- p.3 l.19: typo: appropriate.

- p.7 l.9: typo: $4(10)$

- p.16 l.1: sans, without

- p.16 l.10: missing normal vector.

- p.26 l.17: M.sc. thesis is usually not peer reviewed.

**References**

N. Calonne, C. Geindreau, and F. Flin. Macroscopic modeling for heat and water vapor transfer in dry snow by homogenization. *J Phys Chem B*, 118(47):13393–13403, 2014. doi: 10.1021/jp5052535.

B. R. Pinzer, M. Schneebeli, and T. U. Kaempfer. Vapor flux and recrystallization during dry snow metamorphism under a steady temperature gradient as observed by time-lapse micro-tomography. *The Cryosphere*, 6(5):1141–1155, Oct. 2012. ISSN 1994-0424. doi: 10.5194/tc-6-1141-2012.

L. M. Pismen. Diffusion in porous media of a random structure. *Chemical Engineering Science*, 29(5):1227–1236, May 1974. ISSN 0009-2509. doi: 10.1016/0009-2509(74)80122-3.

R. A. Sommerfeld, I. Friedman, and M. Nilles. The Fractionation of Natural Isotopes During Temperature Gradient Metamorphism of Snow. In H. G. Jones and W. J. Orville-Thomas, editors, *Seasonal Snowcovers: Physics, Chemistry, Hydrology*, NATO ASI Series, pages 95–105. Springer Netherlands, Dordrecht, 1987. ISBN 978-94-009-3947-9. doi: 10.1007/978-94-009-3947-9\_5.

S. Torquato. *Random Heterogeneous Materials: Microstructure and Macroscopic Properties*. Interdisciplinary Applied Mathematics. Springer, 2002.

---

## Referee Comment (RC2) · Anonymous Referee #2 · 30 Oct 2019

**Comments on Manuscript #tc-2019-143:**
*"Revisiting the vapor diffusion coefficient in dry snow"*
*Andrew Hansen*

**General Comments:**
This manuscript proposes to revisit the concept of Apparent Diffusion Coefficient (ADC) in dry snow, which is known to have led to a quite extensive literature, exhibiting a high dispersion in the quantitative results. The author first presents the main studies in a concise, balanced and objective way. He then proposes a definition for the ADC with 4 specific mechanisms that need to be taken into account for its accurate estimation. The author then proposes 3 theoretical models of increasing difficulty to express the ADC in terms of pore and ice volume fractions. He then revisit most of the literature studies, and takes into account their peculiarities to provide ADC estimations that are consistent with those of the proposed definition, showing that ADC values are generally between 1 and 1.3, with a much lower dispersion than that given by the original literature studies.

This manuscript is a sound and interesting contribution to the vapor coefficient literature, and allows better deciphering the high dispersion of the quantitative results observed. In overall, it is clear, didactic and well written, but would benefit from (i) a clearer statement of the main model hypotheses and (ii) additional explanations (e.g. ice blockage exactly compensated by shortened diffusion paths).

Here are some suggestions the author may find useful to improve the manuscript:

1. Objectives and definitions:
To my understanding, the diffusion coefficient the author is interested in is not really an effective diffusion coefficient, but an apparent one: it takes into account mechanisms that are not really based on diffusion processes (e.g. phase changes), but consider them as contributing to the overall diffusion.
To my opinion, the estimation of the ADC is close to an ill-defined problem: I think it would be better not to hide specific processes in an apparent diffusion coefficient, but to address each physical process separately. It is important (i) for a better understanding of the different processes and (ii) for a better modeling of the vapor transfer through the snow cover (modular approach, with possible separate improvements). As the author knows, such a modeling has been e.g. proposed in Calonne et al 2014 and Calonne et al 2015, where specific source terms are used to describe phase changes. Microscale studies of TG metamorphism in controlled conditions (e.g. Pinzer et al, 2012, Calonne et al, 2015, etc.) and appropriate simulations (e.g. Kaempfer and Plapp, 2009) are also very promising means to better understand and identify the different processes, leading to a better quantification of macroscopic vapor transfers in snow.
Given the extensive (and extremely confusing) literature on the vapor diffusion coefficient of snow, I think this manuscript is a sound and interesting contribution to snow science, as it provides simple ways to estimate the order of magnitude of the apparent diffusion coefficient, and, last but not least, clarifies and reconciles most of literature results.
However, I would suggest the following improvements:
-stating more clearly from the start of the manuscript that the diffusion coefficient addressed here is an apparent diffusion coefficient (e.g. in the title, abstract, introduction, definition part...).
-recalling the objectives of ADC estimations, and if adequate, explaining the interest of ADC computations over other approaches (e.g. considering the effective diffusion coefficient and phase change effects separately).

2. Exact compensation of ice blockage by shortened diffusion paths (see p. 17 and 22):
I had some difficulties to understand this point. Adding some explanations would help the reader (see detailed comments).

3. Surface Kinetics:
From the manuscript, I understand the models would work especially well when diffusion through pores is very slow as compared to sublimation/deposition processes. Would the model still be valid if surface kinetics phenomena are of the same order or longer than diffusion times? At least a small paragraph would be welcome on this topic.

4. Discussion of the model parameters and domain of validity of the proposed results.
From the 3 equations (55 to 57) used by the author to model the ADC and from the related figures (Fig. 8-9), it clearly appears that the diffusion enhancement only depends on the volume fraction of ice (1 parameter only). This description has the advantage to be very simple, but raise several questions, e.g.:

    -What about the impact of the ice microstructure (grain and pore size, connectivity, ice shapes, anisotropy…) on the real ADC? E.g., are the results the same for an horizontally (cf Fig. 6) or a vertically oriented layered microstructure of same volume fractions? Have large open pore structures the same ADC than small closed pore structures of the same volume fractions? Formulas have been obtained under strong hypotheses (isotropy, e.g.), but problems related to ADC are inherently linked to snow microstructures obtained under strong TG, i.e., which typically involve the formation of vertically elongated anisotropic structures.

    -What about temperature effects? When temperature decreases (e.g. from -0.5 to -5°C), metamorphism is known to be strongly inhibited while diffusion coefficient and conductivities stay nearly the same (cf e.g. Massmann, 1998 and Calonne et al, 2011). Is the real ADC impacted by temperature and how is it reflected by the proposed formulas?

All these questions are probably difficult to precisely answer in this manuscript but some hints could be given in the discussion.
At least, the main hypotheses of the model should be recalled in the conclusion part.

**Detailed and minor comments:**
p. 2, lines 3-4: *"The net effect is that ice grains act as an instantaneous source and sink of water vapor, thereby shortening the diffusion path of a water molecule"*.
    I would replace "instantaneous" by "quasi-instantaneous" (here, and everywhere in the paper) : depending on the conditions (temperature, facet orientation, etc.), the kinetics at the interface may strongly impede the phase change and impact the resulting geometry. See e.g. Yokoyama, 1990; Libbrecht et al, 2005; Flin and Brzoska, 2008; Libbrecht and Rickerby, 2013, etc.
p. 3, lines 1-2: *"For instance, the numerical studies of Christon (1990), Pinzer et al. (2012), and Calonne et al. (2014) all use different methods to evaluate the mass flux and/or the diffusion coefficient."*
    Actually, as truly explained by the author just a line before, the main problem is probably not the method, but the definition. E.g., in Calonne et al, 2014, we did not consider the sublimation condensation effects (mechanism #2 of the author's definition), as it is not a real diffusion process, but a phase change that can be viewed as contributing to the overall apparent diffusion process. Replacing "methods" by "definitions" might probably be more accurate.
p. 4, lines 15-19: *"Convection is neglected. Convection only occurs in extreme weather conditions such as near the top of a snowpack in the presence of a **strong wind** or extremely large temperature gradients. Foslien (1994) provides support for this assumption through the calculation of a Rayleigh*

*number for porous media. His results show the Rayleigh for snow is an order of magnitude below the number required for the onset of convection..."*

What kind of convection the author actually wants to address here?

   -1) Forced convection as in Calonne et al, 2015 (i.e., wind pumping -with e.g. Reynolds number, etc.)

or

   -2) Natural convection as in Kaempfer et al (2005) (i.e., natural convection, due to the fact that cold air is heavier than warm one -with Rayleigh number)?

As the author know, the physical mechanisms (and associated characteristic numbers) are different for these two distinct processes. It would be better to address these two process in distinct paragraphs.

p. 8, line 15-16: *"Vapor diffusion in snow is driven by temperature gradients and, therefore, it is useful to examine relations between macroscale and microscale temperature gradients."*

This is a small detail, but strictly speaking, the first part of the sentence is not perfectly true, as vapor diffusion in snow can be also caused by curvature effects (see e.g. Brzoska et al, 2008), at least at microscale. Replacing "Vapor diffusion in snow" by "Macroscale vapor diffusion in snow" or "Large scale vapor diffusion in snow" would be more accurate.

p. 6, line 10: "*To account for this altered time scale for water vapor diffusion, the notion of intrinsic time,..."*

For me, the terminology "intrinsic" is unclear in this context. I would suggest to replace "intrinsic" by "apparent", here and everywhere in the text.

p. 16, Fig. 7: *"x-axis: kg m⁻³"*

=> blank spaces to be suppressed.

p. 16, Fig. 7: *"Figure 7. Thermal conductivity analytical prediction of Foslien (1994) versus finite element predictions of Riche and Schneebeli (2013)."*

To my understanding, the comparison made here is a comparison with a very specific fit of the dataset of Riche and Schneebeli (2013), namely the vertical component of the thermal effective conductivity, and for FC and DH only (see Fig 7, p 224: "kz FC and DH").

In addition, I am not sure the red curve of Riche and Schneebeli (2013) can be really considered as "a finite element prediction": for me, it is rather a fit computed from a subset of kz values (FC and DH only), obtained from their whole dataset.

p. 16, line 10: missing brackets or parentheses.

p. 17, lines 14-15: "Preserving the ratio of (ki/kha) is a necessity to preserve the local temperature gradient field obtained from the heat transfer analysis."

At first glance, I found this idea quite difficult to understand. The author should add some explanations (or an example on a specific configuration, with a figure?).

p. 17, lines 26-27: *"Note that the averaging process is done over the humid air volume as opposed to the total volume as the effects of ice blockage and the shortened diffusion paths from hand-to-hand vapor transport cancel each other out."*

Same as just before: some additional explanations would be welcome.

p. 21, lines 20-25: *"Perhaps the most extensive numerical study of mass transport is found in Calonne et al. (2014). They performed finite element analyses for 35 RVE's of snow spanning a density range from 100–500 kgm-3. Unfortunately, it is not possible to reconcile their results with the analytical models presented, as they solved a fundamentally different boundary value problem for mass transfer driven by the relation substitutions shown in (51) where the thermal conductivity of ice was set to zero. In contrast, in the analytical models, the influence of ki is retained to obtain the local temperature gradient field. The influence of ki is then removed in the calculation of the diffusion coefficient."*

I agree with the author. However, it should be mentioned that, in the model of Calonne et al, phase change can be described using specific sources terms (see Calonne et al, 2014 and Calonne et al, 2015).

p. 22, lines 15-16: *"Arguments from stereology, combined with experimental observation, suggest that the influence of ice blocking diffusion paths is canceled out by the shortened diffusion paths from hand to hand vapor transport."*

Same as p. 17, lines 26-27: some additional explanations or justifications would be welcome.

p. 22, lines 26-27: *"Importantly, Eq. (59) represents the mass flux per unit area of snow and not just the flux across the humid air phase as the ice blockage impeding diffusion is exactly countered by the shortened diffusion paths of water molecules"*

Same as just before: some additional explanations or justifications would be welcome.

Minor typo: a "t" is missing in "importantly".

**References**

Brzoska, J.-B., F. Flin and J. Barckicke, 2008. Explicit iterative computation of diffusive vapour field in the 3-D snow matrix : preliminary results for low flux metamorphism, Ann. Glaciol., 48, 13-18, doi: 10.3189/172756408784700798.

Calonne, N., C. Geindreau, F. Flin, 2014. Macroscopic modeling for heat and water vapor transfer in dry snow by homogenization, J. Phys. Chem. B, 118 (47), 13393–13403, doi : 10.1021/jp5052535.

Calonne, N., C. Geindreau, F. Flin, 2015. Macroscopic modeling of heat and water vapor transfer with phase change in dry snow based on an upscaling method : Influence of air convection, J. Geophys. Res. : Earth Surf., 120, 2476-2497, doi : 10.1002/2015JF003605.

Calonne, N., et al., 2015. CellDyM: A room temperature operating cryogenic cell for the dynamic monitoring of snow metamorphism by time-lapse X-ray microtomography, Geophysical Research Letters, 42 (10), 3911–3918.

Flin, F. and J.-B. Brzoska, 2008. The temperature gradient metamorphism of snow : vapour diffusion model and application to tomographic images, Ann. Glaciol., 49, 17-21, doi: 10.3189/172756408787814834.

Foslien, W., 1994. A modern mixture theory applied to heat and mass transfer in snow, M.S. thesis, University of Wyoming, Laramie, WY, USA.

Kaempfer, T. U., and M. Plapp, 2009. Phase-field modeling of dry snow metamorphism,Physical Review E, 79 (3), 031,502.

Kaempfer, T. U., M. Schneebeli, M., and S. A. Sokratov, 2005. A microstructural approach to model heat transfer in snow, Geo. Res. Lett., 32, https://doi.org/10.1029/2005GL023873.

Libbrecht, K. G., 2005. The physics of snow crystals, Reports on Progress in Physics, 68 (4), 855–895, doi: 10.1088/0034-4885/68/4/R03.

Libbrecht, K. G., and M. E. Rickerby, 2013. Measurements of surface attachment kinetics for faceted ice crystal growth, Journal of Crystal Growth, 377, 1–8.

Massman, W. A review of the molecular diffusivities of H2O, CO2, CH4, CO, O3, SO2, NH3, N2O, NO, and NO2 in air, O2 and N2 near STP. Atmos. Environ. 1998, 32, 1111–1127.

Pinzer, B. R., M. Schneebeli, and T. U. Kaempfer, 2012. Vapor flux and recrystallization during dry snow metamorphism under a steady temperature gradient as observed by time-lapse micro-tomography, The Cryosphere, 6, 1141–1155, https://doi.org/10.5194/tc-6-1141-2012.

Riche, F. and M. Schneebeli, 2013. Thermal conductivity of snow measured by three independent methods and anisotropy considerations, The Cryosphere, 7, 217–227, https://doi.org/10.5194/tc-7-217-2013, 2013.

Yokoyama, E., and T. Kuroda, 1990. Pattern formation in growth of snow crystals occurring in the surface kinetic process and the diffusion process, Physical Review A, 41 (4), 2038.

---

## Author Comment (AC2) · 5 Nov 2019

NOTE: For clarity, the author's response is shown in green font.

**General Comments**

I would like to thank the Reviewer for a thoughtful and thorough review of the paper. The comments are very helpful in any effort to improve the work.

I have endeavored to respond to the ***Specific Comments*** and points raised by the Reviewer, and they are addressed later in this response. However, it is the ***General Comments*** that are far more interesting, to me, and worthy of an introductory discussion.

The Reviewer notes: *"The main crux is to assess whether the hand-to-hand vapor transport is physically related to mass flux."* The Reviewer states a bit later that: *"In my opinion the effective diffusion coefficient is a linear response of water vapor and ice transport to a thermal driving force. Since the phase transitions that take place at the microscale serve as a temporal storage of vapor, i.e. ice, it should in principle reduce the effective transport, and therefore reduce the effective diffusion coefficient."*

In the present paper, I have taken a much different perspective than that of the Reviewer—one, I should add, that is not new. Specifically, my position is that hand-to-hand vapor transport is, indeed, physically related to mass flux of water vapor. Moreover, the ice phase should be viewed as a near instantaneous source and sink of water vapor transport, thereby shortening diffusion paths and enhancing diffusion rates. The key attribute of this reasoning is that water vapor molecules are indistinguishable from one another. Hence, water vapor condensing on the bottom of an ice grain is indistinguishable, in form, to water vapor sublimating off the top of an ice grain.

I think the two differing views articulated above can be readily resolved by comments made by Reviewer No. 2 who states:

*"To my understanding, the diffusion coefficient the author is interested in is not really an effective diffusion coefficient, but an apparent one: it takes into account mechanisms that are not really based on diffusion processes (e.g. phase changes), but consider them as contributing to the overall diffusion."*

Reviewer #2 refers to this diffusion coefficient throughout the review as an Apparent Diffusion Coefficient (ADC). I completely agree with her/his description above and I will alter the terminology of the paper to use the word "apparent" when describing the diffusion coefficient. Moreover, the title will be altered to include the wording "apparent vapor diffusion coefficient."

The reason for seeking to quantify the ADC is that its value is precisely what is needed to quantify mass diffusion of water vapor in snow as it relates to the governing field equations of heat and mass transfer for snow. The combined mass balance and energy balance equation, developed in Hansen and Foslien (2015), is given by

$$\left( \phi_{ha}\, \gamma_{ha}\, C_{ha}^V + \phi_i\, \gamma_i\, C_i^V + \mu_{sg}\, \phi_{ha}\, \frac{d\gamma_v}{d\theta} \right) \frac{\partial \theta}{\partial t} = \frac{\partial}{\partial x}\left( k + \mu_{sg} D_s \frac{d\gamma_v}{d\theta} \right) \frac{\partial \theta}{\partial x} \tag{1}$$

The notion of an Apparent Diffusion Coefficient ($D_s$) and the presence of hand-to-hand mass transfer resulting in enhanced diffusion is best demonstrated through an analysis of the mechanics of heat and mass transfer through a layered microstructure of ice and humid air. To be clear, I am not discussing snow at the moment as such a discussion only clouds the salient point of the meaning of the ADC. That said, heat and mass transfer surrounding an ice crust is an important problem in snow metamorphism and the similarities to the layered microstructure are striking.

Consider energy transport of humid air subjected to a temperature gradient. The humid air energy flux developed follows the classic work on *Transport Phenomena* by Bird et al. (1960). The total energy flux for a mixture may be written as

$$q = q^{(c)} + q^{(d)} \ , \tag{2}$$

where $q^{(c)}$ is the conductive flux and $q^{(d)}$ represents "contribution from the interdiffusion of the various species present" (Bird, 1960). Utilizing Fourier's law for conduction and Fick's law for mass diffusion, the energy flux for humid air may be represented as

$$q = -\left( k + \mu_{sg}\, D_{v\text{-}a}\, \frac{d\gamma_v}{d\theta} \right) \nabla \theta \ , \tag{3}$$

where $D_{v\text{-}a}$ is the diffusion coefficient for water vapor in air.

Now consider a layered ice/humid air microstructure subjected to a vertical temperature gradient as shown (Figure 4 in the present paper). We wish to examine the role of mass diffusion on energy transfer.

[Figure]

Figure 4. (a) Layered ice/humid air microstructure (b) Unit cell showing diffusion off the top surface.

The Reviewer's view of the diffusion coefficient is that "*phase transitions that take place at the microscale serve as a temporal storage of vapor, i.e. ice, it should in principle reduce the effective transport, and therefore reduce the effective diffusion coefficient.*" Indeed, for this special microstructure, diffusion is choked off entirely as water vapor molecules cannot freely move through the mixture but rather are deposited on the warmer side of the ice layer. Experiments of Hammonds et al. (2015) confirm this phenomenon as ice deposition occurs on the warmer side of an ice lens while sublimation occurs on the colder side. Hence, in a layered microstructure, individual water molecules are doomed to be trapped within the microstructure, just as the Reviewer posits.

On the other hand, if one begins with the observation that mass transfer through the microstructure shown in the above figure is occurring under the influence of a temperature gradient, one is led to the concept of an ADC. Moreover, a critical mechanism in the ADC is hand-to-hand vapor transport through the ice phase. As water molecules are deposited on one side of the ice, they are simultaneously sublimating off the other side. The net effect is that the ice phase, in this case, acts to enhance diffusion rates by shortening diffusion paths compared to the diffusion of humid air alone.

Under the influence of a macroscale temperature gradient, an energy balance of the layered microstructure requires the energy flux to be the same in the humid air and the ice as well as that of the mixture. Hence, one can write for a 1-D analysis:

$$q = q_i = q_{ha} \; , \tag{4}$$

where subscripts i and ha denote the ice phase, and humid air phase, respectively, and $q$ represents the energy flux of the mixture. The energy fluxes of the ice phase and humid air phase are given by, respectively,

$$q_i = k_i \left(\frac{d\theta}{dx}\right)_i \; , \tag{5}$$

and

$$q_{ha} = \left( k_{ha} + \mu_{sg} D_{v-a} \frac{d\gamma_v}{d\theta} \right) \left(\frac{d\theta}{dx}\right)_{ha} \; . \tag{6}$$

The constituent temperature gradients are related to the temperature gradient of the mixture as

$$\left(\frac{d\theta}{dx}\right) = \phi_i \left(\frac{d\theta}{dx}\right)_i + \phi_{ha} \left(\frac{d\theta}{dx}\right)_{ha} \; , \tag{7}$$

where $\phi_\alpha$ represents the volume fraction of constituent $\alpha$.

The above results may be combined to yield the energy flux for the mixture (macroscale continuum) as

$$q = \left[ \frac{k_i\left(k_{ha} + \mu_{sg} D_{v-a}\frac{d\gamma_v}{d\theta}\right)}{\phi_i\left(k_{ha} + \mu_{sg} D_{v-a}\frac{d\gamma_v}{d\theta}\right) + \phi_{ha} k_i} \right] \left(\frac{d\theta}{dx}\right) \; . \tag{8}$$

It should be emphasized that the above is developed strictly from an energy balance of the microstructure.

One can manipulate Eq. (8) in a few interesting ways. For example, let's eliminate the diffusion term and focus only on conduction of the two-phases. The heat flux then assumes the familiar form given by

$$q = \left[\frac{k_i\ k_{ha}}{\phi_i\ k_{ha} + \phi_{ha}\ k_i}\right]\left(\frac{d\theta}{dx}\right) \ .$$
(9)

The above shows the effective thermal conductivity of the continuum is

$$k = \left[\frac{k_i\ k_{ha}}{\phi_i\ k_{ha} + \phi_{ha}\ k_i}\right] \ .$$
(10)

Returning to Eq. (8), an order of magnitude analysis reveals the first term in the denominator is negligible, leading to

$$q \ = \ \left(\frac{k_{ha}}{\phi_{ha}} + \frac{\mu_{sg}\ D_{v-a}\ \frac{d\gamma_v}{d\theta}}{\phi_{ha}}\right)\left(\frac{d\theta}{dx}\right) \ .$$
(11)

Importantly, the diffusion coefficient associated with latent heat transfer is given by $D_{v\text{-}a}/\phi_{ha}$. Equation (11) indicates mass diffusion of water vapor is occurring throughout the continuous medium of the mixture and is enhanced compared to humid air alone. Finally, one can write the energy flux of the mixture as

$$q = -\left(k + \mu_{sg}\ D_s\ \frac{d\gamma_v}{d\theta}\right)\left(\frac{d\theta}{dx}\right) \ .$$
(12)

Equating (11) and (12) we arrive at an expression for the ADC given by

$$D_s = \frac{D_{v-a}}{\phi_{ha}} \ .$$
(13)

The above equation for the ADC is precisely that which is predicted by the hand-to-hand diffusion analogy described in the paper. In brief, diffusion paths are shortened by the ice phase and the apparent diffusion is enhanced.

I will revise the introductory of a second draft to provide a clear explanation of the desire to compute an ADC while also better articulating the differences in definitions of the diffusion coefficient. My hope is that the motivations of the article will be much improved while substantial clarity and framing of prior research is also accomplished.

**Anonymous Referee #1**

**1 General comments**
This paper addresses a long standing and vividly debated topic in the snow science community and makes a contribution by unifying the definition of effective diffusion. The paper converts the wide variety of predicted and measured effective diffusive coefficient Ds to reasonable range. The paper starts from a mixture theory representation of transport of mass and energy in snow and defines an upper bound for the effective vapor diffusion coefficient as a function the density of snow. By introducing an objective definition of vapor transport one can compare several previous experimental studies on effective vapor

transport. The paper shows that all measurements are well below the upper bound. In addition, it formulates two models based on diffusion with and without tortuosity.

I agree with the author that effective vapor diffusion is a mind-boggling topic and can become confusing from time to time. The author attempts to make a first step in clearing the clutter by starting from the first principles of mixture theory. He identifies 4 ways the diffusion path of water can be altered with respect to pure vapor diffusion without an ice matrix interfering. 1) Blockage of diffusion paths by ice, 2) Shortened diffusion paths due to the phase transition at the interface. 3) Enhanced local diffusion by difference in heat conduction in both phases. 4) Tortuosity, lengthening diffusion paths. The author states that any model for the effective diffusion coefficient these 4 principles should be accounted for.

The author introduces an artificial velocity of the vapor molecules that is calculated by the path length that it travels divided by the time spend in the vapor phase only, effectively deleting the time it spends in the ice phase. Hereby the definition of the effective diffusion coefficient is enhanced. Based on this concept, the author defines an upper bound for Ds, which is a density correction based on this ice volume fraction.

The first model, which is a superposition of a layered and a tubular microstructure, leads to a slightly different density correction. The last model that should include tortuosity and is based on the method developed by Calonne et al. (2014). They calculate the effective diffusion by changing transport coefficients from the heat conductivity problem in terms of the transport of the vapor flux coefficients. Here the transport in the ice phase was set to 0. In this work the ice transport coefficient is set to 100 times that of vapor, to implement the idea of shortened pathways or intrinsic/artificial velocity.

I have two comments related to the above paragraph. First, the opening statement is a bit misleading in that the model referenced is not that of: *"a superposition of a layered and a tubular microstructure."* A much better view of the situation is that both the upper bound and the model for 1-D vapor transport lead to an identical equation governing the ADC given by:

$$D_{\mathrm{s}}\left(\frac{d\theta}{dx}\right) = D_{\mathrm{v-a}}\left(\frac{d\theta}{dx}\right)_{\mathrm{ha}} .$$
(14)

Given a macroscale temperature gradient (TG), the difference in the upper bound model and the 1-D model arises entirely from the different humid air temperature gradients in the respective models. For the layered microstructure, the humid air TG is known, leading to the upper bound. For the 1-D model, the TG in the humid air phase is unknown, requiring some approximation. In the interest of a reasonable solution, the thermal conductivity model of Hansen and Foslien (2015) is utilized to estimate the humid air TG. To describe the use of the thermal conductivity analysis of the humid air TG as "the model" is simply inaccurate. Other approaches to estimating the TG of the humid air are welcome but the functional form of Eq. (14) is unaltered.

The last sentence of this paragraph states: *"In this work the ice transport coefficient is set to 100 times that of vapor, to implement the idea of shortened pathways or intrinsic/artificial velocity."* While this statement is true in that—by setting the ice coefficient to 100 times that of vapor, it implements the idea of shortened pathways—the real reason this was done was to reproduce the TG vector field of the thermal problem. The shortening of diffusion pathways or intrinsic/artificial velocity is an artifact (albeit a nice physical interpretation) of the desire to reproduce the temperature gradient field for the mass diffusion problem.

My main concern is on the introduction of the intrinsic time or intrinsic/artificial velocity. It is introduced to delete the time the vapor molecules spend in the ice phase, yielding an effective higher velocity to the

vapor molecules, since the distance is that it apparently travelled through the ice phase is still counted. If this approach of apparent mass transfer is taken, the apparent movement of the ice matrix in opposite direction of the flux should be accounted for as well. The latter is important, because without it the mass balance of your snowpack wouldn't be complete. The intrinsic time is confusing and as concluded in the end of the paper is not really necessary. However, it is still the basis to compute the effective diffusion coefficient, which raises the question if the meaning of this definition is physical or not.

The notion of an Apparent Diffusion Coefficient to describe water vapor diffusion in snow is not new. Yosida (1955) is the first to clearly articulate this concept and Pinzer et al. (2012) devote an entire section of their article to _numerous_ studies embracing this view. The use of an intrinsic velocity, perhaps better described by Reviewer #2 as an "apparent" velocity, is a means to provide a mathematical model of the hand-to-hand diffusion phenomenon. The diffusion example provided in the introductory remarks clearly shows the ADC has physical meaning—that is, it is the coefficient needed to model heat and mass transfer at the macroscale.

Regarding the apparent movement of the ice matrix in the opposite direction of the flux, while true, this motion is completely negligible in the context of estimating the velocity of mass diffusion of water vapor. The time scales for thermal equilibrium of an RVE are on the order of minutes while time scales for movement of the ice matrix is on the order of days or weeks. Pinzer et al. (2012) report growth rates of ice due to metamorphism to be on the order of $100~\mu m~d^{-1}$—the ice motion is simply inconsequential.

The main crux is to assess whether the hand-to-hand vapor transport is physically related to mass flux. Since the water molecules diffusivity in ice is orders of magnitudes slower giving it a large of infinite average velocity is not physical. Mechanisms to which actual transport is being enhanced is if the local vapor flux in the pore space is on average more enhanced, than the reductions that are expected. It is known that tortuosity $\tau$ is leading to an upper bound for $D_s = (1 - \phi_i)D_{v-a}/\tau$ (Pismen, 1974). Note that this is only a reduction of $D_{v-a}$. A heterogeneous distribution of temperature gradients in porous media will lead to actual locally enhanced vapor flux. Bounds for these fields could be found/developed based on e.g. Torquato (2002), but as far as I know, cannot exceed a density correction factor, as the author suggests for his upper bound.

My interpretation of the given models is that they use the linearization between the vapor concentration gradient and the temperature gradient at the microscale and find density corrections for the latter given 1) a layered, 2) a mix between tubular and layered microstructures. The physical interpretation of the last model is not clear to me, especially how tortuosity is included. If the models are restated in the context of enhanced temperature gradients, they might be more physical/useful. In conclusion, the debate around the effective diffusion coefficient is in need for a uniform and clear definition. The approach, which is chosen in this paper, is interesting, but also highly confusing. In my opinion the effective diffusion coefficient is a linear response of water vapor and ice transport to a thermal driving force. Since the phase transitions that take place at the microscale serve as a temporal storage of vapor, i.e. ice, it should in principle reduce the effective transport, and therefore reduce the effective diffusion coefficient. Given that it is the authors choice to define it in a different manner, I will respond with some specific comments below.

The discussion in the above two paragraphs has been thoroughly addressed in my opening remarks. I am attempting to quantify the ADC that is needed for the continuum equations of heat and mass transfer. The ADC has a very different physical meaning than the diffusion coefficient articulated by the Reviewer.

**2 Specific comments**

2.1 Section 1: History

- General: Introduction is quite sparse regarding the explanation of the different mechanisms for enhanced vapor diffusion. For understanding the introduction can be more detailed. example:

    The introduction of the paper will be revised in an effort to address comments by the Reviewer.

- p.2. l.6: More explanation of this experiment of Sommerfeld et al. (1987) would clarify how this was measured, which assumptions are made etc. same for the studies Pinzer et al. (2012) and Calonne et al. (2014)

    It is reasonable to start a revised draft introduction by pointing out the different definitions used in describing water vapor diffusion. Doing so, should alleviate the main concern of the Reviewer. However, it is not so easy to discuss the various assumptions of prior studies without detailing out the diffusion mechanisms discussed in Section 2.2. It would be far too premature to try and introduce these in an introduction to the paper.

- p.2 l.32: What is physical difference between blockage of diffusion paths and tortuosity?

    A nice way to view ice blockage is simply the volume occupied by the ice precludes mass diffusion in that volume. Hence, in any volume average calculation, the blockage reduces the apparent diffusion coefficient.

    A clear understanding that blockage is not tortuosity might be viewed in the context of the columnar pore microstructure—see Figure 6(a). The ice phase clearly acts as a blockage to diffusion, yet there is no tortuosity in such a microstructure.

    In the case of snow, the volume of the ice phase again precludes diffusion from occurring in this region of an RVE. Tortuosity, resulting in longer diffusion paths for water vapor, further reduces the apparent diffusion coefficient. In brief, ice blockage and tortuosity are different mechanisms for retarding diffusion rates.

- General: From the history or the introduction, it should become clear to the reader which studies have been performed and where the discrepancies lie. To me it is not clear without reading the actual referenced papers what has been measured, modeled or simulated. Suggestion: start with stating that differences in studies are mainly based on the assumptions of contributing mechanisms and the lack of a consistent definition, and depart from there by describing which assumptions they made.

    Please see my discussion above of the second comment in this Section as it addresses the Reviewer's points made here.

2.2 Section 2: Mass transfer in snow

- p.5. l.8: If isotropy can be relaxed please discuss it somewhere at a later stage.

The diffusion coefficient model developed here-in relies on isotropic thermal conductivity as developed by Foslien (1994). Anisotropy may enter the development through the influence of the humid air temperature gradient $(d\theta/dx)_{ha}$ in Eq. (14) of this response (Eq. 38 of the paper). Below are some of my thoughts on the topic of anisotropy although the topic warrants a thorough investigation.

During kinetic metamorphism, recrystallization causes the formation of thicker vertical-oriented ice structures aligned with the TG. In rough terms, the structure is trending toward a columnar pore microstructure of Figure 6(a) of the paper. As a result, the thermal conductivity is rising, as the columnar pore microstructure represents an upper bound on thermal conductivity for a given density.

In the case of diffusion, the opposite is true in that the columnar pore microstructure retards apparent diffusion compared to a general snow microstructure. Hence, the apparent diffusion is expected to be lower in the vertical TG direction than in the horizontal direction. I'll note this description is precisely the opposite of what is shown in Calonne et al. (2014). This contrast is directly attributable to the different definition of the diffusion coefficient used by Calonne (2014) versus the Apparent Diffusion Coefficient studied here. Fundamentally different diffusion mechanisms are at play in the respective models.

Relative to diffusion in the horizontal direction, my experience with studying elastic properties of composite materials suggests that the structural changes of kinetic metamorphism in the vertical direction will have minimal effect on the horizontal diffusion coefficient.

In summary, I expect kinetic metamorphism caused by a vertical temperature gradient will lower the diffusion coefficient in the vertical direction and have minimal effect on the horizontal diffusion coefficient.

To quantify anisotropic diffusion effects, one needs to evaluate the humid air TG for snow exhibiting kinetic crystal growth. The most obvious means of obtaining this information is through finite element simulations of the thermal conductivity problem. Pinzer (2012) achieved this for 3 snow samples imaged over time under a temperature gradient. One of those samples shows diffusion decreasing with time while the other two are rather steady in time.

A second draft will provide a discussion of anisotropy.

- p.6. l.11: The introduction of apparent time is confusing to me. If we were to couple measured diffusion to mass transport, we would need to add the residence time spend in the crystal to the time travelled in humid air? If you make the connection to residence time, one could 'track' single molecules.

  This concern has been addressed through my discussion of the Apparent Diffusion Coefficient. The ADC is the coefficient needed for continuum formulations of heat and mass transfer. A critical feature of computing the ADC is the introduction of intrinsic/apparent time.

- p.7. l.15: With eq.11 the definition of Ds is given, which is basically the effective linear response to the physical driving force. Maybe consider this as a natural bridge to describe what it is effected by given the physics at the microscale.

I'm not sure what is being asked or stated here. I would be happy to address this topic with additional clarification.

- p.7. l.20: In this paragraph we supposed to be convinced that it is useful to frame Ds in connection to the balance equations. We know that they are dependent on Ds but a quick calculation of the relative importance of the second term is less than 5 ‰.

  The suggestion that the relative significance of $D_s$ in latent heat transfer is a means of dismissing the definition of $D_s$ is simply not valid. Moreover, $D_s$ plays the dominant role in the calculation of deposition or sublimation in mass transfer—see Hansen and Foslien (2015) for a discussion of the "mass supply."

  It is not only useful to frame $D_s$ in connection with the balance equations, determining the correct value of $D_s$ to be used in the balance equations is the absolute essence of this paper. In addition to providing a precise definition of $D_s$, (the ADC), the paper shows that there is not nearly as much scatter in its value as perhaps perceived in the snow community today.

  Finally, the Reviewer's suggestion that the relative contribution of latent heat transfer is 5% seems a bit arbitrary. Hansen and Foslien (2015) show the contribution of latent heat to the energy flux may range anywhere from near 0 to as high as 30%—high temperatures and lower densities increase the importance of latent heat transfer. I'll note the predictions of Hansen and Foslien fall in line with the experimental values of Hammonds et al. (2015) who determine the latent heat transfer to be between 8 and 10% for the densities and temperatures in their paper.

- p.8. l.9: Despite the title, there is no formal definition of Ds = . . .. One could use eq.(11) but then it means that is connected to the flux of snow, which can only be understood in terms of mass flux.

  Eq. (11) provides a precise definition of the meaning of $D_s$, the Apparent Diffusion Coefficient. Its use allows one to quantify the apparent movement of water vapor through snow. I believe my introductory comments in this response have addressed this point.

- p.8. Fig.3: This figure is confusing, suggesting that the bonds are colder than the grains, by the color bar. The RVE is also on the small side, usually one expects the system size to be roughly 10 times the element (grain) size.

  This comment is an important observation on the part of the Reviewer—indeed, there is confusion and it is caused by the color bar. The grain bonds are definitely not colder than the grains. In the original figure, produced in the 1990s, the grains were shown in red and the bonds in blue to help clearly identify these regions. This figure was produced before the era of the CT scan and, at the time, it was our attempt to simulate a snow microstructure. We wanted to distinguish grains from bonds, hence the different colors of red and blue. I'll note the figure is not readily revised, given its age.

  The purpose of the figure, presently, is to show ice blockage in the volume of the RVE caused by occupation of the ice phase as well as the effects of ice blockage on a surface— shown in gold. I believe the figure does a nice job of showing the blockage. As to the size of an RVE, this particular figure is not intended for computational purposes. The smaller size is simply a visual aid.

I believe the Reviewer's confusion is eliminated by removing the color bar and simply showing a line for the temperature gradient. In addition, text may be added to the caption of a revised draft to clarify the situation.

I appreciate the Reviewer's observations on the original figure.

- p.9. l.15: By taking out $dv/d\theta$ of the integral, you make quite some assumptions, i.e. the temperature gradient and the vapor gradient are locally always reacting the same, regardless of its surroundings. In principle one should solve the Laplace equation for your vapor density, given the appropriate boundary conditions, which can include the interface dynamics of the phase transition. Stating clearly which assumptions you make is in place here.

  The Reviewer is raising two different points here. First, $dv/d\theta$ may be removed from the integral as the temperature changes within an RVE are extremely small. This term is essentially constant throughout the RVE.

  The latter point raised by the Reviewer is a legitimate concern. A complete coupled heat and mass transfer analysis, a nontrivial endeavor, incorporating interface dynamics and phase transition is really the correct approach to a complete solution. In that vein, to my knowledge, the only work to accomplish this is the pioneering work of Christon et al. (1990, 1994). As noted in their work, the result of their analysis produces an enhanced diffusion coefficient over that of humid air at all densities.

  I'll also remark the work of Christon et al. has, in the past, been the subject of criticism (unfairly in my view), owing to the simpler microstructures they were forced to use. The CT scan for snow and the awesome computing power needed for such an analysis had yet to arrive. That said, my experience is that volume averaging to determine thermomechanical properties is remarkably forgiving when it comes to defining a precise microstructure. My guess is that Christon's results would hold up very well today when compared to a more sophisticated analysis using a snow RVE microstructure.

  I have provided substantial discussion of the interface boundary conditions in the first comment of Section 2.3 below. Some discussion of the assumptions required for these boundary conditions will be added to any revised draft of the paper.

- p.9. l.22: Suggesting linking macro to micro scale by explicitly relating js to jv, which gives you an expression for Ds that shows by what physics it is influenced.

  I am confused by this statement. Again, I'm happy to attempt a response given some additional clarity here.

- p.10 l.1: I believe 1) and 4) are in principle the same. From a microstructural perspective I would suggest that there are two mechanisms: 1) Tortuosity that increases the free path length of water vapor molecules. and 2) Phase transitions that, driven by temperature gradients at the ice-air interface, change the boundary conditions for the mass transfer of the vapor concentration and therefore alters the diffusion paths in a structural manner.

  Diffusion mechanisms 1) and 4) are not the same. I addressed this issue in the third comment in Section 2.1 comments above.

The Reviewer's comments about diffusion mechanisms are really a reflection of her/his view of diffusion versus the Apparent Diffusion Coefficient I am computing. What is needed in the paper is a clear distinction between these views. Using the terminology ADC in a revised draft should achieve this objective.

- p.10. l.9: My general thought is that it would not be correct to use $v_v$, since actual molecules aren't travelling through the ice phase with (in)finite speed, and if one looks at it from an apparent perspective, one has to account for the apparent movement of the ice crystals in the opposite direction that is represented by the ice interface velocity in opposite direction.

  Using $v_v$, in conjunction with the hand-to-hand analogy of water vapor transport, is entirely appropriate for computing the ADC. The rigorous 1-D energy balance for the layered microstructure presented earlier is a fine demonstration of this approach. Again, the central point is the differing views on the definition of diffusion. The use of the term Apparent Diffusion Coefficient should resolve this confusion.

  I previously addressed the issue of the apparent movement of ice crystals in the opposite direction. The velocity of such motion is completely negligible compared to the diffusion velocity in humid air.

2.3 Section 3: Models

- p.14 l.4 & Fig 5: The argument that blockage and shortening of diffusion paths are equally frequent is based on the frequency of the ice phase, which is, for isotropic media the same. Given that the chance for shortening or blocking is given by underlying physics one still need to solve the vapor diffusion equation including the correct boundary conditions. Therefore, the relative chances can still be very different.

  The boundary conditions for the vapor diffusion equation at the microscale are well established and have been consistently applied by researchers studying vapor diffusion in snow (Christon 1990, 1994, Flinn and Brzoska 2008, Pinzer et al., 2012, and others). Assuming local thermodynamic equilibrium at the ice/humid air interface leads to interface temperatures that are the same in ice and humid air with equal and opposite energy flux vectors.

  The mass diffusion problem requires specification of the mass concentration, normally through specification of the vapor pressure. Equilibrium is invoked and the equilibrium vapor pressure over a flat surface is utilized on the ice/humid air interface. There are 3 possibilities one must account for that could vary the saturated vapor pressure boundary condition.

  1. Kelvin's law shows that the equilibrium vapor pressure over curved surfaces exceeds that of flat surfaces. Christon (1994) and Flanner and Zender (2006) show the correction for a curved surface only becomes significant for radii of curvature below $(10)^{-7} \, m^{-1}$. Hence, the ice surface in a seasonal snow cover may be considered flat relative to the Kelvin effect. Curvature effects are of importance in freshly fallen snow.

  2. Latent heat at deposition sites may cause local heating at the ice/humid air interface. However, heating does not occur due to the extremely high thermal conductivity of the ice phase relative to the humid air. The ice immediately conducts the heat away.

3. Surface kinetics may lead to a build-up of the mass concentration at the ice/humid air interface leading to a supersaturation of the vapor pressure. Hobbs (1974) provides an expression for the condensation of water vapor to ice driven by the difference in the vapor pressure and the saturated vapor pressure of ice ($p - p^{sat}$). Hansen and Foslien (2015) show the time for this difference to dissipate to 0.1% of the original difference is on the order of $(10)^{-3}$ seconds in snow. Hence, the vapor pressure in a pore can be assumed to be the saturated vapor pressure given the time scale of interest for diffusion may be hours or days.

Note that this discussion does not involve any specific assumption as to the definition of the diffusion coefficient. It is simply driven by the physics of heat and mass transfer at the microscale.

- p.15. l.1: Is the underlying assumption that the microstructure is a superposition of a layered and pore structures?

  The underlying assumption is not that the microstructure is a superposition of layered and pore microstructures.

  I think an important point has been missed regarding the pore and layered microstructures. A persistent theme, here and in other venues, is that these microstructures do not represent the complex porous medium of snow and therefore the Foslien 1994 model is called into question.

  It is important to look at the role of these microstructures in a much different way than that of some microstructural representation of snow. Specifically, the microstructures support heat transfer *mechanisms* in the humid air occurring in parallel with ice or in series with ice. Both of these *mechanisms* are on constant display at the microscale within a complex snow RVE. For example, given an ice grain in a vertical temperature gradient field, heat transfer in a pore above the grain appears in a serial behavior while the pore space beside a grain sees more parallel heat transfer similar to the columnar pore microstructure. Foslien (1994) astutely introduced stereology to capture the significance of these mechanisms. The merit of Foslien's (1994) thermal conductivity model is well represented by the predictions of thermal conductivity seen in Figure 7 (revised to also include work of Calonne et al., 2011). The predictions made decades in advance of finite element analyses RVE's developed from CT scans are quite remarkable.

- p.15. l.4: the naming Pore, is highly confusing. I would suggest something like tubular microstructure.

  I agree that the use of the word "pore" alone is confusing. All occurrences of "pore" in this context will be changed to "columnar pore." Pore is descriptive in the sense that it brings to mind the humid air phase. Columnar is an indicator of a long prismatic structure. The two words together should bring to mind the proper picture.

- p.15. l.7: There is only a slight difference in figure 7, but there is no information on how well this fit is doing to predict real conductivities.

  The Reviewer's point here is well taken. In a revised draft, language will be added to the paper to discuss the relative accuracies of the curve fit to the finite element predictions of Riche and Schneebeli (2013).

I have also altered Figure 7 to include a second set of predictions by Foslien (1994) benchmarked against the work of Calonne et al. (2011). While not quite as close, the curves are reasonably accurate for lower density snow, and, where deviations occur at the higher densities, the model of Foslien tends to the known thermal conductivity of ice.

I want to reiterate that I believe too much emphasis is focused on the origins of Foslien's thermal conductivity model. It is clear the model is capturing a reasonable estimate of the thermal conductivity and I believe the model also captures the key heat transfer mechanisms at play. Of course, anyone is free to propose a different approach to computing the humid air TG, including detailed finite element analyses of RVE's. I do not believe the results will be notably different—it is simply not possible given the curves of Figure 7.

[Figure]

**Figure 7—revised**. Thermal conductivity analytical predictions of Foslien (1994) versus curve fits of finite element predictions of Riche and Schneebeli (2013) and Calonne et al. (2011).

- p.16. Fig. 7: What is the purpose of this figure? In the caption you write finite element predictions, but this should be a fit to finite element simulations.

  The purpose of Figure 7 is to provide the reader with some confidence in the thermal conductivity model of Foslien. The importance of this model is that it allows one to estimate the TG of the humid air. Of course, there may be other avenues to estimate or rigorously compute the humid TG. The approach I took was readily available in an analytical form, and further, is well grounded to available models of thermal conductivity.

  The caption will be altered to reflect the curves of Riche and Schneebeli (2013) and Calonne et al. (2011) are curve fits to finite element predictions.

- p.17. l.16: This is just not physical in terms of the [Calonne.2014] paper.

  I agree with the Reviewer's comment relative to the Calonne 2014 paper. However, in terms of the pursuit of the Apparent Diffusion Coefficient in this paper, the physical interpretation is entirely defensible. The goal is to reproduce the TG vector field in the diffusion problem because the local TG vector field drives local (microscale) diffusion.

- p.18. l.6: This equation is not derived but stated? It is hard to follow this line of thought without understanding the appendix of Calonne et al. (2014)

  Eq. (54), identical to Eq. (57), was derived by referring to Eq. (43) along with the appropriate substitutions from thermal conductivity to diffusion. The approach essentially relies on volume averaging the TG gradient vector field in the humid air phase as it drives the local diffusion velocity.

  Greater clarity is needed, and additional detail will be provided on this topic in a second draft.

- p.18. l.7: How is tortuosity exactly included? Tortuosity is generally described by the path that molecules travel through a porous media without altered paths due to phase transitions, which is basically governed by laminar flow in an inert media. How is this relevant or included in this model?

  The point raised here is important as to the language of the paper. I will alter the name of the noted model from "Tortuosity Model" to "Curvilinear Diffusion Path Model." The distance of a curvilinear diffusion path falls between the straight line 1-D model and that of a tortuosity path described by the Reviewer.

- p.19. Fig.8: Redundant figure.

  Figure 8 will be removed and all references to Figures 8 & 9 will be combined into a single figure.

- p.20. l.10: There are 6 values plotted, but I can't retrace them to measurements from the [Pinzer.2012] paper, a little more information could be helpful on how you did this, which figures or data you actually used.

  The values were taken from max/min estimates of the finite element predictions of vapor flux in Figure 11 of Pinzer (2012). These predictions are very stable with time and Pinzer et al. (2012) state that the finite element predictions should be considered more reliable than the particle image velocimetry measurements.

2.4 Section 4: Conclusion

- As the author states Eq. (59) should be the vapor mass flux in snow. There is no need for an introduction of shortened diffusion paths, or intrinsic velocity, because it is not physical. In the conclusion it is stated that in principle the definition of Ds is the same as eq.59. This to me is confusing because you use $v_v$ in eq.(11). Maybe I'm missing the general point of the paper, but one way or the other this should be clarified. Figures: In general the figures are not illustrative to the paper. Fig 3 is very confusing, suggesting bonds are colder than grains. My suggestion is to

replace Fig 1,2,3 with a slice of a 3D image of actual snow and indicate the different processes leading to vapor diffusion (tortuosity and phase transitions at the interface)

> The general point of the paper is to determine reasonable estimates of the Apparent Diffusion Coefficient needed for a continuum formulation of macroscale heat and mass transfer in snow. Moreover, the paper shows that the variability in prior estimates of this parameter is significantly reduced when all diffusion mechanisms have been accounted for.
>
> To compute an ADC, there is a need for introducing shortened diffusion paths—hand-to-hand mass transport—and this transport mechanism is rigorously demonstrated in the layered microstructure of ice and humid air. The connection to snow follows directly when stereology is incorporated into the analysis, i.e., ice crystals shorten diffusion paths while also blocking diffusion.
>
> I personally think Figures 1-3 are important, recognizing the very important changes to Figure 3 noted previously. Figure 1 is beneficial to framing the analysis of snow as a mixture of ice and humid air. Figure 2 helps to clarify the meanings of the heat flux and mass (water vapor) flux crossing a surface in snow. That said, Figure 2 could be eliminated as I feel it is the least important of the 3 figures.
>
> Figure 3 is useful to show ice blockage (red and blue) in the context of volume or ice blockage (gold) across a surface. In my opinion, replacing Figure 3 by a real RVE of a CT scan, while more contemporary, does not add to the purpose of the figure. Of course, this change can be done, although I personally do not have ownership of any such CT scans.
>
> In a revised draft of the article, the introduction will be rewritten in an effort to address the concerns of the Reviewer regarding the general point of the article. A discussion of the differing definitions of the diffusion coefficient will also be provided along with motivations for seeking to quantify the Apparent Diffusion Coefficient.

**3 Technical comments**

- Notation: The author decides on using $\theta$ as a symbol for temperature and $\xi$ as symbol for coordinates. This might be the convention in mixture theory, but for readability and adaptation to the snow community I would advise on T and x respectively.

> There are compelling reasons to retain $\xi$ as a symbol of coordinates at the microscale (the scale of the RVE). The coordinate $x$ is reserved for the macroscale and should be distinguished from its microscale counterpart, $\xi$. Naming them the same is truly inviting confusion.
>
> As for using $\theta$ to describe temperature, I think it is a bit unfair to describe this as the convention of mixture theory. Numerous texts on continuum mechanics, with no discussion of mixtures, use $\theta$ for a symbol for temperature. In some cases, T is reserved for the first Piola-Kirchhoff stress tensor—an important stress measure for finite deformation, and snow certainly sees finite deformations when loaded. Furthermore, the paper of Hansen and Foslien (2015)—*The Cryosphere* develops the governing equations for heat and mass transfer in snow utilizing $\theta$ to represent temperature. Finally, T appears as a subscript in the present work to represent the total length across a RVE.
>
> I would prefer to leave the notation as is currently written. However, I am not married to the notion of using $\theta$ to represent temperature. I'll leave that to the purview of the editor.

- Use of the word 'may' in e.g. p.7.13 sounds to me as if the followed expression is lucrative in the sense that we may also use something else. The word 'can' in this context would be more appropriate, however, it is a matter of taste.

 *The word "may" has been replaced by "can".*

- p.3 l.19: typo: appropriate.

 *Typo corrected—thank you*

- p.7 l.9: typo: 4(10)

 *The number $4(10)^{-3}$ has been altered to read $4 \times 10^{-3}$ in the text.*

- p.16 l.1: sans, without

 *The word "sans" has been replaced by "without".*

- p.16 l.10: missing normal vector.

 *Normal vector added—thank you*

- p.26 l.17: M.sc. thesis is usually not peer reviewed.

 *I'm not sure what the point of this comment is. Some of the important results in the present paper may only be found in the thesis of Foslien (1994). If there is a question as to peer review, Hansen and Foslien (2015)—The Cryosphere is a de facto rigorous review of Foslien's thesis. The thesis is both a rigorous and complete study of the subject.*

**References**

N. Calonne, C. Geindreau, and F. Flin. Macroscopic modeling for heat and water vapor transfer in dry snow by homogenization. J Phys Chem B, 118(47):13393–13403, 2014. doi: 10.1021/jp5052535.

B. R. Pinzer, M. Schneebeli, and T. U. Kaempfer. Vapor flux and recrystallization during dry snow metamorphism under a steady temperature gradient as observed by time-lapse microtomography. The Cryosphere, 6(5):1141–1155, Oct. 2012. ISSN 1994-0424. doi: 10.5194/tc-6-1141-2012.

L. M. Pismen. Diffusion in porous media of a random structure. Chemical Engineering Science, 29(5):1227–1236, May 1974. ISSN 0009-2509. doi: 10.1016/0009-2509(74)80122-3.

R. A. Sommerfeld, I. Friedman, and M. Nilles. The Fractionation of Natural Isotopes During Temperature Gradient Metamorphism of Snow. In H. G. Jones and W. J. Orville-Thomas, editors, Seasonal Snowcovers: Physics, Chemistry, Hydrology, NATO ASI Series, pages 95–105. Springer Netherlands, Dordrecht, 1987. ISBN 978-94-009-3947-9. doi: 10.1007/978-94-009-3947-9\_5.

S. Torquato. Random Heterogeneous Materials: Microstructure and Macroscopic Properties.

Interdisciplinary Applied Mathematics. Springer, 2002.

**References**

Bird, R. B., Stewart, W. E., and Lightfoot, E.: Transport Phenomena, John Wiley and Sons, New York, 1960.

Calonne, N., Flin, F., Morin, S., Lesaffre, 5 B., Roscoat, R., and Geindreau, C.: Numerical and experimental investigations of the effective thermal conductivity of snow, Geophys. Res. Lett., 38, L23501, https://doi.org/10.1029/2011GL049234, 2011.

Calonne, N., Geindreau, C., and Flin, F.: Macroscopic modeling for heat and water vapor transport in dry snow by homogenization, J. Phys. Chem. B, 118, 13393-13403, https://doi.org/10.10221/jp5052535, 2014.

Christon, M.: 3-D transient microanalysis of multi-phase heat and mass transfer in ice lattices, PhD thesis, Colorado State University, Fort
10 Collins, CO, USA, 1990.

Christon, M., Burns, P., and Sommerfeld, R.: Quasi-steady temperature gradient metamorphism in idealized, dry snow, Numer. Heat Tr. A-Appl., 25, 259–278, https://doi.org/10.1080/10407789408955948, 1994.

Flanner,M. G. and Zender, C. S.: Linking snowpack microphysics and albedo evolution, J. Geophys. Res., 111, D12208, doi:10.1029/2005JD006834, 2006.

Flin, F. and Brzoska, J.: The temperature-gradient metamorphism of snow: vapour diffusion model and application to tomographic images, Ann. Glaciol., 49, 17–21, 2008.

Foslien, W.: A modern mixture theory applied to heat and mass transfer in snow, M.S. thesis, University of Wyoming, Laramie, WY, USA,1994.

Hammonds, K., Lieb-Lappen, R., Baker, I., and Wang, X.J.: Investigating the thermophysical properties of the ice-snow interface under a controlled temperature gradient Part I: Experiments and observations, Cold Reg. Sci. Technol., 120, 135-148, https://doi.org/10.1016/j.coldregions.2015.09.006, 2015.

Hansen, A. C. and Foslien, W.: A macroscale mixture theory analysis of deposition and sublimation rates during heat and mass transfer in dry snow, The Cryosphere, 9, 1857–1878, https://doi.org/10.5194/tc-9-1857-2015, 2015.

Hobbs, P.: Ice Physics, Clarendon Press, Oxford, 1974.

Pinzer, B. R., Schneebeli, M., and Kaempfer, T. U.: Vapor flux and recrystallization during dry snow metamorphism under a steady temperature gradient as observed by time-lapse micro-tomography, The Cryosphere, 6, 1141–1155, https://doi.org/10.5194/tc-6-1141-2012, 2012.

Riche, F. and Schneebeli, M.: Thermal conductivity of snow measured by three independent methods and anisotropy considerations, The Cryosphere, 7, 217–227, https://doi.org/10.5194/tc-7-217-2013, 2013.

---

## Author Comment (AC3) · 2 Dec 2019

Please find my response to reviewer comments in the supplement. Thank you.

Please also note the supplement to this comment:
https://www.the-cryosphere-discuss.net/tc-2019-143/tc-2019-143-AC3-supplement.pdf
* * *